# The Actual Clinical Situation Ruthlessly Exposes the Challenge of Rational Care for Nosocomial and Community-Acquired Infections and Requires Even More Efforts for Satisfactory Antibiotic Stewardship

**DOI:** 10.3390/antibiotics14060561

**Published:** 2025-05-30

**Authors:** Hans H. Diebner, A. Melina Wallrafen, Nina Timmesfeld, Tim Rahmel, Hartmuth Nowak

**Affiliations:** 1Abteilung für Medizinische Informatik, Biometrie und Epidemiologie, Ruhr-Universität Bochum, D-44780 Bochum, Germany; melina.wallrafen@rub.de (A.M.W.); nina.timmesfeld@rub.de (N.T.); 2Klinik für Anästhesiologie, Intensivmedizin und Schmerztherapie, Universitätsklinikum Knappschaftskrankenhaus Bochum, In der Schornau 23-25, D-44892 Bochum, Germany; tim.rahmel@knappschaft-kliniken.de (T.R.); hartmuth.nowak@knappschaft-kliniken.de (H.N.); 3Zentrum für Künstliche Intelligenz, Medizininformatik und Datenwissenschaften, Universitätsklinikum Knappschaftskrankenhaus Bochum, In der Schornau 23-25, D-44892 Bochum, Germany

**Keywords:** antibiotic stewardship, antimicrobial resistance, nosocomial infections, community-acquired infections, intensive care unit, cross-resistance

## Abstract

**Background:** Antimicrobial resistance is one of the 10 most pressing health problems worldwide. **Methods:** First steps toward harnessing the complex dynamics of antibiotic resistance are presented. To accomplish this, we first shift down a gear and try to understand the actual driving dynamics behind the development of resistance in a specific clinical department. Analyses are based on the clinical and microbiological data of a German hospital over an observation period of more than 7 years, which we evaluate descriptively and semi-quantitatively in order to obtain a basis for informed and intelligent action in terms of antibiotic stewardship. **Results:** The specific results include the observed increase in the resistance rate with increasing overall consumption, while increases over time independent of consumption are fairly moderate. Vancocymin and refoximin are an exception in the development of resistance, as resistance to these substances appears to decrease with increasing consumption. However, there have been substantial dose adjustments for these substances, which are likely to be decisive here. An intra-host increase in resistance due to treatment time on the one hand and repeated treatments on the other is observed. Within the sub-cohort of ineffectively treated patients, i.e., with resistance to the antibiotic, mortality increases on average, but with ampicillin/sulbactam as a striking exception. Patients with infections caused by ampicillin-resistant bacteria have a lower mortality rate. The observed resistance rates of the eight most frequently administered antibiotics show a temporal variability that includes random fluctuations as well as decidedly regular cycles. The time series associated with the various antibiotics show pairwise time lag correlations, which indicates the existence of retardedly mediated cross-resistance. **Conclusions:** We conclude with an outlook on upcoming further analyses and a draft action plan on how to control and harness the complex dynamics observed by means of successful, informed, and intelligent antibiotic stewardship.

## 1. Introduction

For critically ill patients in the intensive care unit with severe bacterial infections, the rapid and targeted initiation of adequate antibiotic therapy is of utmost importance for the patient’s outcome. Antibiotic resistance has an increasingly important influence on the effectiveness of therapy and the chances of clinical recovery. The current worldwide increase in antibiotic-resistant bacteria has therefore been defined by the World Health Organization (WHO) as one of the 10 greatest global health threats to humanity, which requires decisive action through differentiated measures [1].

Antibiotic resistance is mainly caused by overuse or misuse of antibiotics. Alternative causes as differences in subpopulations are also currently discussed [2]. The choice of substance used plays the major role here, but dosage aspects and the resulting low antibiotic levels in the blood or the target organ system are also relevant, as this can lead to the selection of resistant subpopulations of pathogens under antibiotic therapy. The appropriate dosage of antibiotics is therefore not only important for patient outcome, but also for the development of resistance itself. In addition to choosing the individual substance for the patient, the physician must therefore keep an eye on the sufficient dosage and also the so-called “resistance pressure” that they automatically exert on the individual patient but also on the entire cohort of the corresponding ward and hospital through the choice of antibiotic.

As a consequence, structured optimization measures against antibiotic resistance are of utmost medical importance. Specifically, in addition to the pharmacological development of new substances/antibiotic classes by the pharmaceutical industry, the rational and responsible use of antibiotics is particularly important. This preventive approach to avoiding resistance is subsumed under the term antibiotic stewardship (ABS) [3].

The aim of ABS is to treat patients in the best possible way and at the same time prevent selection processes and resistance from occurring in the bacteria. To this end, it would be desirable to regularly evaluate and optimize the rational use of antimicrobial substances with an integrative approach that combines routine intensive care and microbiological data from the clinic with mathematical and methodological analyses: a clinical decision support system. To establish such a system, however, the above-mentioned hurdles arising from the complexity of the clinical situation must be overcome. Reflection is now required and the natural next step is to first better understand the given complexity.

Even though Germany still has relatively low incidences of resistant bacterial pathogens compared with other (Southern and Eastern) European countries, the trend in this country is also increasing alarmingly, particularly with certain Gram-negative pathogens [4]. In contrast to Gram-positive pathogens, this group of multi-resistant Gram-negative bacteria poses the greatest challenge, as several hundred different resistances with a pleiotropic pool of different resistance mechanisms have already been identified. Frequent basic principles of antibiotic resistance are the inactivation of *β*-lactam antibiotics by means of chemical cleavage by so-called *β*-lactamases, as well as the reduction of the antibiotic in the bacterial periplasm through the expression of efflux pumps or the loss of porins. In particular, the large number of very different *β*-lactamases influences the efficacy of the largest antibiotic class of *β*-lactams. Special *β*-lactamases (so-called carbapenemases) can give rise to the fact that no *β*-lactam antibiotic, including the group of carbapenems as reserve substances, still has clinical efficacy against corresponding pathogens [5]. However, multi-resistance also affects structurally and functionally unrelated antimicrobial active substances, as will be discussed again below in the context of cross-resistance.

Major innovations in the development of antibiotics for Gram-negative pathogens therefore primarily consist of the combination of a “new” *β*-lactamase inhibitor with an existing substance. However, there is already resistance to these “new” antibiotics. On the one hand, *β*-lactamase inhibitors are only effective against certain *β*-lactamases. On the other hand, their use can lead to the selection of subpopulations of the treated pathogen that are resistant to this combination of active substances [6]. The use of new antibiotics is therefore strictly limited to cases in which other substances are no longer effective and will not be able to solve the antibiotic resistance problem in the long term.

Preventive concepts such as ABS are therefore of crucial importance in preventing the development of resistance. In addition to the actual microbiological identification of the pathogen, this includes in particular the selection of the appropriate antibiotic, including pharmacokinetic and pharmacodynamic aspects. Clinically, the efficacy of a substance is determined by measuring the minimum inhibitory concentration (MIC). The MIC is the lowest effective concentration of an antibiotic that still prevents the replication of a pathogen in a culture. Clinical threshold values defined for the respective pathogens (in Europe by the EUCAST—“European Committee on Antimicrobial Susceptibility Testing”) then identify a pathogen as resistant (R) or susceptible (S) to the respective antibiotic tested and support clinicians in selecting the correct substance in the form of the so-called antibiogram [7].

Regarding resistance development, however, the aspect of the active substance concentration achieved in the target compartment is also of decisive importance. It is currently assumed that a pathogen population that triggers an infection has different subpopulations with different MICs [8]. It is therefore assumed that an antibiotic tested sensitive per se leads to the selection of subpopulations of the treated pathogen with a higher MIC if blood or target organ levels are inadequate, resulting in the clinical manifestation of resistance. This development can occur either within a patient (e.g., in the case of recurrent antibiotic episodes during long courses of intensive care), but also across patient boundaries in the entire ward or hospital cohort over a longer period of time, as there is inevitably also an exchange of the microbiome between patients (direct and indirect/iatrogenic) [9].

As a countermeasure, attempts are made by ensuring sufficiently high antibiotic levels and thus a sufficient dosage. This has recently been accomplished, for example, by determining the blood or tissue levels of a substance as part of therapeutic drug monitoring (TDM) and any resulting dose adjustments [10,11]. However, a third category “I” in susceptibility testing (English for “susceptible for increased exposure”) also addresses the requirement for sufficient antibiotic levels, which clinically means that the substance has been tested as sensitive but requires an increased dosage. With regard to the overall cohort, the local resistance situation in the form of ward and/or hospital-specific resistance statistics also plays a relevant role.

Another possible ABS strategy proposed and analyzed by numerous authors is the cyclical allocation of antibiotics or so-called mixing (other terms are in circulation). These are strategies at hospitals with the aim of contributing to a reduction in the prevalence of resistant germs by means of spatial (between departments, mixing) or cyclical (temporal) variations in the proportions of consumption of different antibiotic groups. An in-depth discussion of the difference between the cyclical strategies can be found in [12]. According to a recent systematic review [13], there is only one randomized controlled trial (RCT) on the effectiveness of such strategies. However, there are some studies that at least compared systematic cyclical administration strategies and standard administration between clinics or carried out before-and-after comparisons (cross-over) [13,14].

Meta-analyses report a small effect (on average), with individual studies reporting a clinically relevant effect [13,14]. In the RCT study, cycling performed worse than the control ABS [13]. In numerous studies, “mixing” was chosen as the control strategy, i.e., an alternating exchange between departments [12]. Cycling, i.e., a temporal periodic change simultaneously across all departments, showed no difference to the mixing procedure, which is not surprising from a theoretical point of view if one assumes sufficiently isolated conditions between the departments. Nevertheless, all studies (RCT and cohort studies) including the cross-over studies were considered together in the meta-analyses, regardless of whether they tested against “mixing” or “without strategy”, which must be viewed very critically and neglects the important separate consideration of mixing and cycling. However, in the meta-analysis by [14], a distinction was made between the two control strategies as part of a secondary analysis. In Gram-positive bacteria, the cycling strategy showed a slightly stronger effect in terms of avoiding resistance. It appears that there is no general evaluation independent of other biological and medical boundary conditions. It is therefore possible that the cycling concept itself is not well thought out. The question of whether cycling or mixing contributes to the “rational and responsible use of antibiotics” has therefore not been conclusively clarified.

“Scheduled cycling” according to a fixed scheme means that the informed, i.e., rational use of antibiotics, is deliberately avoided so that conceptually a control or zero strategy rather than a verum strategy is defined here. In a seminal cycling study published by [15], the authors investigated interrupted time series in the administration of aminoclycosides. These irregular antibiotic prescriptions followed an observed pattern in the emergence of resistant germs and by no means a fixed periodic cycling scheme. In retrospect, it appears that the authors intuitively used the method that is now referred to as “clinical cycling” and which is actually superior to scheduled cycling because it is based on clinical evidence for the necessity of a switching strategy. Unsurprisingly, scheduled cycling strategies have not yet found any clinical application in practice. Moreover, cycling is explicitly not recommended in a German S3 guideline [16]. Noteworthily, the validity already expired on 31 January 2024, so an update has been overdue for more than a year, i.e., the S3 guideline does not currently provide any valid orientation for ABS.

The latest insights into the genesis of cross-resistance and delay phenomena in this genesis make the strictly timed cyclical variation appear particularly naive in retrospect [17,18,19]. Cross-resistance can arise due to genetic changes in the bacterial strain. They can acquire resistance via gene transfer—via conjugation or transformation—from other bacteria. Transmission via bacteriophages (transduction) is also possible (e.g., [20]). However, it has also been clear for some time that multi-resistance must actually be interpreted as cross-resistance per se, as multidrug resistance is a term that refers to mechanisms of resistance by chromosomal genes. In this case, as [17] points out, an exposure to a single drug leads to cross-resistance to many other structurally and functionally unrelated drugs. As mentioned above, an important mechanism identified for multidrug resistance in bacteria is drug efflux by membrane transporters.

Results such as those by [21] show that current strategies for the use of critical antibiotics are not sufficient to avoid unexpected antibiotic cross-resistance. They found that cross-resistance to daptomycin occurred in vancomycin-resistant *Enterococcus faecium* when rifaximin was used. The resulting resistance formation has been shown not only for antibiotic use, but also for antiviral drugs: bacteria exposed in vitro to antiviral drugs with antibacterial properties can develop multiple resistance mutations associated with cross-resistance to antibiotics [21].

In another cross-correlation study, it was shown that considering the previous consumption as well as the incidence density of strains during the previous quarter proved to be the best model to explain carbapenem resistance of *P. aeruginosa* strains based on meropenem consumption during a given quarter [22]. Furthermore, a correlation was found between antibiotic consumption and the occurrence of multidrug-resistant organisms [23]. Also worth mentioning is the meta-analysis on the effect of combination therapies on the development of resistance, which shows mixed results. In some cases, the prognosis is better with combinations, but in others, the opposite effect is seen [24].

Arguably, despite the aforementioned counterarguments, the poor performance of “scheduled cycling” does not speak against cycling per se, but rather against cycling that is not carried out intelligently. One could even say that conceptually, scheduled cycling corresponds more to a “placebo-like” (control) ABS, whereas clinical cycling based on clinical expertise corresponds to the verum group. In the context of this interpretation, the many studies on cycling are in fact evaluation procedures instead for assessing whether the respective “clinical cycling” is viable on the basis of implicit clinical knowledge in comparison with a non-informed cycling strategy. In other words, despite the clinically contra-indicated settings of scheduled cycling programs, these strategies represent a kind of basic model structure whose quantitative explanation is the basis for the description of more complex switching strategies (clinical cycling).

In our own preliminary work [25], we have created and published a mathematical framework that is suitable for adequately quantifying the effect of clinical cycling and, in borderline cases, scheduled cycling. It is worth mentioning that in most of the studies conducted, no quantification of the degree of mixing or cyclic variation was used. An exception is the study by [26], in which an antibiotic heterogeneity index (AHI) was used. However, it turns out that AHI is completely unsuitable for the intended task. It is a global measure that is invariant to swapping the antibiotic classes. This means that if the consumption shares of two antibiotics are swapped, the AHI remains unchanged.

We therefore developed a mathematical method to correctly calculate temporal and spatial heterogeneities in the consumption of antibiotics and in the prevalence of pathogens. Subsequent correlation analyses revealed a relationship between the heterogeneity of antibiotic consumption and the prevalence of resistant pathogens, indicating a reduction in the prevalence of resistant germs. The heterogeneity changes on the pathogen side follow the changes on the consumption side.

Diversity measures, which are frequently used in ecological studies in particular, were used to calculate “mixing”, i.e., the degree of heterogeneity. These measures are related to entropies, which are used in information theory and especially in statistical physics for the quantitative description of mixing processes (dispersions) and similar phenomena. There is a larger class of such entropies or diversities, whereby the concrete choice depends on the specific problem. However, the Shannon entropy known from information theory is, like AHI, a global entropy, i.e., invariant to permutations of the species. Local heterogeneity measures must therefore be used in order to be able to record changes over time. The Kullback–Leibler entropy is such a local entropy and has proven its worth in our preliminary work in the context of ABS as well as spatio-temporal epidemic patterns [25,27]. However, the concrete choice of the final form of the analysis algorithms depends on specific conditions. The estimated values of some entropies from a larger class of theoretically allowed diversity measures are inaccurate when very small consumption fractions are involved.

It must be mentioned that adaptive model-based optimization in everyday clinical practice is not readily possible on the basis of our preliminary work. Theoretically, it would be conceivable to extrapolate the observed mixing states, i.e., to create possible interaction scenarios, in such a way that an “optimal cycling regime” is achieved, but this extrapolation would take place without any “constraints”, such as those imposed by biological and clinical framework conditions and regulations. However, we note that the foundation for such an optimization has been laid by our preliminary work, also taking into account clinical constraints (antibiogram, pharmacokinetics, and microbiological parameters guidelines) [10,11,25,27,28].

As mentioned, the clinical requirements of ABS also include careful and appropriate microbiological diagnostics. A comprehensive prevalence and mortality study recently addressed which pathogens are the biggest problem children [29]. In summary, in 2019, approximately 14% of all deaths were due to a bacterial infection, based on the 33 most important pathogens. Furthermore, 56% of sepsis-associated deaths died as a result of these 33 infections. It is also noteworthy that 55% of deaths from the 33 bacterial species were due to infections of
*Staphylococcus aureus*;*Escherichia coli*;*Streptococcus pneumoniae*;*Klebsiella pneumoniae*;*Pseudomonas aeruginosa*.Previously, the six pathogens from the so-called ESKAPE series
*Enterococcus faecium*;*Staphylococcus aureus*;*Klebsiella pneumoniae*;*Acinetobacter baumannii*;*Pseudomonas aeruginosa*;*Enterobacter*;were discussed as particularly critical with the highest clinical relevance [30]. It almost goes without saying that successful ABS must keep an eye on the prevalence of these pathogens, which are classified as particularly dangerous, especially with regard to the development of resistance. In this context, we also refer to the studies by [31,32,33], which specifically address mortality due to antibiotic resistance in ICUs.

The obvious preparatory step for such a clinical decision support system in the form of an adaptive optimization is therefore the precise descriptive representation of the clinical conditions and requirements, i.e., the representation of the local resistance situation in the form of resistance statistics. A first attempt towards such statistics will be performed in the present work for the intensive care unit of a German hospital, located in Bochum.

## 2. Results

### 2.1. Basic Summary Statistics

We start our endeavor to better understand the relationships and dynamics of antimicrobial resistance development and its clinical implications with some obvious descriptive evaluations. Simple correlations and summary tables are presented, sometimes using univariable linear regressions, which are not intended here to suggest causality. A retrospective study of this kind cannot and will not do more than highlight correlations in order to ultimately generate hypotheses.

#### 2.1.1. Demographic Data

In addition to sex and age, the demographic data also include the length of stay in the hospital and ICU, as well as the discharge destination and discharge diagnosis. With the exception of the n = 1127 unique ICD10 discharge diagnosis codes, summary Table 1 shows the corresponding frequencies stratified by sex, together with *p*-values from univariable tests. Despite the small *p*-values resulting from the large sample size, hardly any sex differences are discernible, with the exception of a slightly higher age among women.

After merging the demographic data with the dataset containing the antibiotic treatments, it is possible to determine for each distinct sequence of stays per patient whether or not an antibiotic was administered during these sequences and generate a corresponding binary flag. Summary Table 2 shows the demographic characteristics stratified by this AB treatment flag. Male patients are slightly more likely to be treated with antibiotics. Age is not a determinant of being treated or not. As was to be expected due to the additional vulnerability caused by nosocomial or community-acquired infections, death is a more likely destination among the antimicrobially treated patients, whereby AB treatment is conceived as surrogate for infection. In addition, the duration of hospitalization as well as the length of stay at the ICU are significantly longer for the AB-treated cohort. Of note, the subject entity of Table 1 and Table 2 are ward stay rather than an individual person.

#### 2.1.2. Antibiotic Treatment Sequences

A series of uninterrupted daily antibiotic treatments per patient is regarded as a treatment sequence with an associated duration. If there are at least 2 days without treatment, the subsequent treatments are assigned to new sequences. The number of treatment sequences per patient exhibits the frequencies depicted in Table 3. Apparently, one patient had 10 treatment sequences, but most (n = 3222) had only one sequence. The durations of the treatment sequences obey the distribution (density curve) depicted in Figure 1. Fifteen sequences have durations of at least 50 days, but most sequences (n = 2777) are shorter than 5 days.

#### 2.1.3. Multiple Simultaneous or Immediately Consecutive Antimicrobial Treatments

A total of 526 patients were administered more than five different antimicrobial substances during their hospital stay. In 92 people, more than 10 substances were administered during their stay. Table 4 shows the number of occurrences of $n_{A}B$ antimicrobials administered simultaneously per patient per day. For example, one patient was administered 10 agents simultaneously on a given day. However, on most patient days, 14,419 patient days to be precise, only one substance was administered, but on about as many patient days, at least two were administered. Even at this early stage, we are almost overwhelmed by the prevailing complexity, to put it somewhat drastically. Which substance should we switch to if all of them are already being administered simultaneously in order to maintain the therapeutic goal?

#### 2.1.4. Antibiogram

The antibiogram data contain 155,594 exposures of isolated pathogens to antibiotics. The majority yielded susceptibility, and very few antibiotic challenges did not lead to any usable results. See frequency Table 5 for a complete summary. The 10 most frequently isolated pathogens are shown in Table 6.

Skipping the two opportunistic pathogenic yeast species, *Candida albicans* and *Candida glabrata*, we identify the most dangerous and problematic bacteria from the ESKAPE series and those associated with high mortality rate [29,30] and, in addition, *Staphylococcus epidermidis*. Thus, in terms of pathogen prevalence, the clinic studied shows the known global characteristics.

In total, isolates were taken from 41 different sample types. However, some were unspecific, e.g., smear, and some were more differentiated, e.g., ulcer smear. Sample types with frequencies above 1000 each are with decreasing frequency: tracheal secretion, urine, anaerobic blood culture, aerobic blood culture, deep swab, and bronchial lavage. Evaluations with sample type as a predictor are not considered in this descriptive review.

#### 2.1.5. Correlation Between Treatment Duration and Resistance

For each treatment sequence, the proportion of all sensitivity tests performed for which resistance was identified is now determined for all isolated pathogens. A correlation is then estimated between the log durations of the sequences and the respective proportions of resistances. We also added the number of antibiotics administered per day as a predictor. Panel A of Figure 2 shows a scatter plot of proportions of resistance versus the logarithms of durations together with a regression line. The occurrence of resistance slightly increases with the duration of treatment. The significant regression result is shown in Table 7 and corresponds to the expected trend, although the effect strength is not very pronounced. For the sake of completeness, Pearson’s correlation coefficient *ρ* is estimated to be significant with *ρ* = 0.066.

#### 2.1.6. Correlation Between ICU Length of Stay and Resistance

Some patients had distinct ICU stays, which are summed up for the following correlation analysis. A moderate correlation between length of ICU stay and the proportion of antimicrobial resistance, $p_{R}$, can be observed as shown in Figure 2B, confirmed by means of a linear regression of $p_{R}$ onto the logarithm of the length of stay (Table 8). Finally, the Pearson’s correlation analysis yields a significant results with the estimated coefficient being *ρ* = 0.128.

In the sense of a consistency check, it remains to show the correlation between the duration of treatment and the length of stay in the ICU. As can be seen in Figure 2C, the corresponding scatter diagram is naturally bounded upwards by the bisector treatment duration = ICU length of stay and is similar in shape to a cone.

Up to this point, a relevant increase in resistance over the course of treatment or ICU stays could not be demonstrated. However, there is a small, formally significant upward trend, and it goes without saying that the development of resistance within individual courses has not been systematically tested. Moreover, because no causality can be proven, the initial presence of resistant germs, which then leads to an extension of the treatment period, cannot be ruled out. However, the intra-host development of resistant germs remains a serious possibility. We will return to this topic in connection with treatment repetitions.

#### 2.1.7. Correlation Between Discharge Destination and the Proportion of Ineffective Antimicrobial Treatment

With regard to the three possible hospital discharge destinations, which are death, home, and external hospital, the question arises as to whether the proportion of antibiotic resistance is a possible risk factor for mortality. A linear regression of the proportion of ineffective antimicrobial treatment on the three-level discharge destination is used here to simplify the analysis and is interpreted as a reverse regression. The negative result observed (see Table 9) may come as something of a surprise, but in fact, the discharge destination does not differ at all in terms of being a possible resistance risk factor. As shown in Table 3, treatment with antibiotics is associated with increased mortality, but antibiotic resistance does not appear to contribute to an additional increase in risk. For the ICU, whose data are analyzed here, this is certainly a relieving result. However, it must be added that we have taken a highly aggregated view of the incidence of resistance here. We will provide antibiotic-specific analyses and take up the issue of mortality later on, which will reveal considerable heterogeneity among the substances.

#### 2.1.8. Correlation Between the Number of Intra-Individual Treatment Sequences and the Proportion of Ineffective Antimicrobial Treatment

Using the result of the calculation of the number of intra-individual antibiotic treatment sequences presented in Table 3 as an independent variable in a linear regression results in a significant increase in the proportion of ineffective antimicrobial treatment, i.e., the proportion of resistance within the performed microbiological sensitivity tests of the given patient. The regression result is shown in Table 10, yielding a considerable increase by roughly 4% per additional AB treatment sequence. However, a causal interpretation should be strongly discouraged.

### 2.2. Consumption Density Time Series of Selected Antibiotics

The nine most frequently administered antibiotics are listed in Table 11. The second column contains the cumulative days of use, while the third column contains the number of individuals who received the corresponding antibiotic.

For these frequently administered antibiotics, with the exception of flucloxacillin, the time series of the proportions of the consumption density of each antibiotic for a given resistance of at least one treated pathogen to this antibiotic relative to the total consumption density of the antibiotic, $p_{A_{R}}\left(t\right)$ (cf. Equation (Equation 1)), are shown in Figure 3, one antibiotic per panel. Flucloxacillin has been skipped as no antibiograms were created for this drug. Caspofungin is an antimycotic, but the time series is shown for the sake of completeness. However, it should be noted that caspofungin is excluded from some of the following analyses. In any case, resistance does not appear to be a pressing problem for caspofungin. Each panel header in Figure 3 contains the reference to the antibiotic that is depicted. In addition, the temporal average as well as the standard deviation of each share of consumption is printed within the corresponding plot as well as a simple fitted linear model ($p_{A_{R}}$ by time) to visualize the time trend (red line with confidence band). All plots have the same y-axis range in order to simplify comparison.

#### 2.2.1. Piperacillin/Tazobactam

Piperacillin is a *β*-lactam antibiotic that belongs to the acylaminopenicillin group and has a very broad spectrum of activity. It is therefore not surprising that piperacillin is also the most frequently administered antibiotic in the ICU studied here. The pure form of piperacillin is *β*-lactamase-labile, which is why the antibiotic is usually combined with a *β*-lactamase inhibitor, usually tazobactam (patent 1993, cf. PubChem). Only this variant was administered in the ICU analyzed.

Piperacillin inhibits bacterial cell wall synthesis and is effective against Gram-negative bacteria and Gram-positive cocci; however, it is primarily used in infections with Gram-negative pathogens. It is therefore not surprising that 6380 samples of isolated Gram-negative bacteria were tested for resistance to piperacillin in the present study, but only 87 Gram-positive pathogens. It should be noted that a total of 9153 antibiograms with Gram-negative, 13,417 with Gram-positive, and 43 with fungi were prepared as part of the study. Incidentally, some samples were also tested for resistance to piperacillin without tazobactam, although it was never prescribed.

The share of ineffectively administered piperacillin as defined in (Equation 1) has a temporal mean of 25% and standard deviation of 5% and shows an only moderate secular drift (visualized by fitting a linear model to the time series) over the roughly 7 years of observation time (see upper left panel of Figure 3). However, it is not only random fluctuation that contributes to the variance, but apparently also systematic cyclical variations. A spectral analysis confirms the presence of an annual period (possibly a seasonal variation) and, in addition, a weakly pronounced 3-year cycle (see Figure 4). With respect to overall consumption, these cycles are only slightly pronounced (cf. right panel of Figure 4) and can thus largely be attributed to the resistance component. A possible attribution to seasonal fluctuations in infection activity, for example, or an effect of non-systematic clinical cycling remains as speculation for the time being. However, it is advisable to keep these phenomena in mind when striving to optimize ABS.

A linear regression of $p_{piperacillin_{R}}\left(t\right)$ on time with the total consumption as an additional covariate used for adjustment yields a significant yet moderate secular drift (*β* = 0.002) and a remarkable increase in the proportion of ineffective piperacillin usage (due to resistance) with increasing total consumption (*β* = 0.023; see Table 12). Thus, the amount of total piperacillin consumption appears to be a driver of resistance. We mention in passing, without going into detail, that the estimation of an interaction between time and consumption results in a positive regression coefficient, with the individual effects becoming slightly negative. Further details would suggest a precision that cannot be achieved in the context of such an exploratory study. Nevertheless, it is obvious that the temporal increase in the resistance fraction is in fact triggered by a temporal increase in consumption. Consequences of this result need to be discussed later in the light of corresponding results for the other antimicrobial substances.

To complete the picture of piperacillin administration, the time course of the average dose per application is also shown. The upper left panel of Figure 5 shows a clear systematic change, albeit with a moderate amplitude. About halfway through the observation period, there is a marked drop in the dose administered, but shortly afterwards, it rises again just as abruptly, briefly to a level above the time average, which is shown by means of a linear regression line.

Of note, although piperacillin has been administered exclusively in combination with the *β*-lactamase inhibitor tazobactam, it was used without addition as a challenge in microbiological tests (see Table 13 for a summary table of the performed piperacillin tests stratified by observed sensitivity). In most cases, however, an antibiogram was performed in parallel for both substances per test unit, and only in a few cases exclusively for piperacillin without Tazobactam, as can be seen from Table 14. The most common pathogens tested for piperacillin (with and w/o tazobactam) resistance stratifed by sensitivity are listed in Table 15 using a cutoff of *n* > 300 for the selection. From the set of pathogens classified as particularly dangerous, i.e., primarily the ESKAPE series, the three Gram-negative bacteria *Escherichia coli*, *Klebsiella pneumoniae*, and *Pseudomonas aeruginosa* are represented here. Table 16 shows the frequencies of the tested sensitivities of the three most important pathogens with respect to piperacillin/tazobactam only.

Finally, and perhaps most relevant, is the observation that mortality does not correlate with resistance, as can be seen in Table 17. However, the slightly higher mortality within the resistance class meets the expectation. Strikingly, both ICU and hospital duration are significantly increased within the resistance stratum. Thereby, only those unique hospitalizations of all patients (noting that there are separate multiple hospitalizations of some patients) in which piperacillin/tazobactam therapy was administered were used to create this table. The laboratory data were also reduced to the subset of piperacillin challenges, and it was determined whether at least one piperacillin resistance of the isolated germs was present per relevant patient. The summary table with sensitivity classes as strata is obtained by merging these datasets.

#### 2.2.2. Meropenem

Meropenem (patent 1986, cf. PubChem), like piperacillin, is a broad-spectrum antibiotic and belongs to the carbapenem class. Meropenem is stable against serine-based *β*-lactamases and therefore has built-in *β*-lactamase protection, so to speak, which must be ensured with the additive tazobactam in the case of piperacillin. Despite stability against *β*-lactamases, meropenem can be combined with the *β*-lactamase inhibitor vaborbactam, which is in use for the treatment of complicated urinary tract infections. However, because this substance combination has only been recently approved, it was not available in the past and has not been applied at the ICU studied.

The clinical picture for meropenem is comparable with that for piperacillin. The proportion of ineffective consumption oscillates around a temporal average of 0.13 with a standard deviation 0.09, as depicted in the upper right panel of Figure 3. Fitting a simple linear model to the time series suggests a mild secular decrease. Again, annual recurrent waves are directly apparent from the time series plot, which is why we refrain from performing an explicit spectral analysis.

Adding the total consumption of meropenem as covariate to a linear regression yields the result depicted in Table 18. The moderate secular decrease already observed from a simple linear regression shown in Figure 3 can be confirmed after the adjustment (*β* = −0.024). Total consumption unfolds a significant and relatively strong impact with *β* = 0.083 units per year. A closer look at the data reveals that a moderate decrease in total consumption over time is responsible for the apparent time effect. As for piperacillin, the effect of total consumption on the share of resistance is therefore also dominant here.

Furthermore, similar to the increasing dose per application of piperacillin, the dose of meropenem is also increasing over time, which is in line with TDM and the resulting dose adjustments [10,11]. The corresponding time course is shown in the middle panel of the first row of Figure 5. About halfway through the observation period, there is a marked increase in the dose administered, but shortly afterwards, it falls again just as abruptly roughly to the baseline dose. The overall trend is shown by means of a linear regression line.

The pronounced wave lasting approx. 1 to 2 years from the beginning of the observation period deserves attention. Whether this is an effect of the change in the definition basis of the sensitivity category *I* and the associated application recommendation remains as speculation. However, because *I* plays a subordinate role with meropenem anyway, this is unlikely. What is more likely is an interaction with a dynamic change in consumption diversity in relation to all antibiotics involved.

Summary Table 19 presents the frequencies of performed meropenem tests stratified by observed sensitivity, and Table 20 presents the frequencies of the most common pathogens tested for meropenem resistance, again grouped by sensitivity. In addition to the piperacillin case, *Enterobacter cloacae* has noteworthy prevalence within the scope of meropenem challenges.

Finally, repeating the above observation again for the piperacillin case, the sensitivity of pathogens with respect to meropenem only has a moderate effect on mortality, as can be seen from Table 21. In this case, however, the increased mortality in the resistance class is slightly more pronounced than for the piperacillin case. Both ICU as well as hospital durations are strikingly strongly prolonged within the resistance stratum.

#### 2.2.3. Caspofungin

Infections caused by *Candida* sp. are also an increasing problem, particularly in nosocomial infections. Due to the focus on antibiotics, we would like to keep the comments on the relatively frequently used antimycotic caspofungin brief at this point. Table 22, which contains the frequencies of the observed sensitivity classes of all pathogens challenged with caspofungin, shows that resistance to caspofungin is not a pressing problem. The left panel in row 2 in Figure 3 also illustrates that resistance was apparently only present for a short time at the beginning of the observation period. Due to the side effect profile and unfavorable interactions with other therapeutic agents, substitution with rezafungin in particular is being considered [34]. However, the ongoing adaptation and change processes must be kept in mind with regard to ABS optimization strategies.

#### 2.2.4. Vancomycin

Vancomycin (patent 1955, cf. PubChem), on the market for quite a long time, is a reserve antibiotic preferably used for the treatment of Gram-positive bacteria. In fact, most applications of vancomycin in the ICU studied here are infections caused by methicillin-resistant *Staphylococcus aureus*, coagulase-negative staphylococci (e.g., Staphylococcus epidermis), or ampicillin-resistant *Enterococcus* sp. The produced antibiograms (results listed in Table 23) show that resistance to vancomycin is so far a serious problem only with respect to *Enterococus faecium* at the studied ICU.

The time course of the proportion of ineffective treatments with vancomycin, $p_{vancomycin_{R}}\left(t\right)$, is depicted in the right panel of the second row of Figure 3 along with a fitted line from a simple linear regression pointing to an increasing secular trend (red line). Estimated mean and standard deviation turn out to be 0.12 and 0.07, respectively. Adding the total consumption as an adjusting covariate to the linear regression gives the estimated coefficients listed in Table 24. The increase in time can be confirmed (*β* = 0.013). Surprisingly, the proportion of ineffective treatments significantly decreases with increasing total consumption (*β* = −0.031). In this case, an interaction term gives a zero effect, thus confirming the somewhat unexpected behavior. The systematic decrease in the administered dose per application is striking. This applies to both the intravenous and the oral form of application, as can be seen in Figure 5 (second row, middle, and right). Whether this is a purely random coincidence or a functional correlation with the decrease in resistance remains speculative for the time being. Also, the apparent escalating oscillation of $p_{vancomycin_{R}}\left(t\right)$ (cf. Figure 3) deserves to be kept in mind urgently.

Table 25 summarizes the demographic/clinical characteristics of vancomycin-treated patients stratified by sensitivity class. Once more, an effect of sensitivity of pathogens with respect to vancomycin on mortality cannot be inferred. However, a *p*-value of 0.07 gives cause for warning. The duration of hospitalization is considerably prolonged, whereas the ICU length of stay is slightly shorter. At this point, however, it should be mentioned that length of stay is conditionally dependent on death. Thus, collinearity of mortality and ICU duration may be the reason for the drop in the length of stay. Unfortunately, numerous treatments with vancocymin took place without an antibiogram being performed. This “untested” sensitivity class results in the highest mortality, followed by the resistance class.

Of note, one patient, who suffered from an acute spontaneous bacterial peritonitis and eventually died, received vancomycin for a duration of 70 days. Moreover, one of the isolated pathogens of this patient, *Pseudomonas aeruginosa*, has been labeled six times as 4MRGN, i.e., as Gram-negative bacterium multi-resistant to four antimicrobial substance classes. It is suspected that vancomycin was used as a last resort in this and many other observed similar cases due to this multi-resistance and co-infections with aggressive Gram-positive bacteria.

In most cases, several active substances were administered simultaneously. The list of the 23 antimicrobials that were given simultaneously with vancomycin during a patient’s hospitalisation with the most different antimicrobial treatments is shown in Table 26. For this patient, on average, five different substances were administered per day. This also applies to the antibiotics already discussed above, albeit for significantly fewer patients and with fewer simultaneously adminstered substances. At this point, we would like to point out that the clinical approach to antimicrobial treatments has a degree of complexity that should not be underestimated and thus illustrates once again how difficult evidence-based ABS actually is. It cannot even be ruled out that the performance based on existing clinical expertise is already very close to optimum. Potential further improvements will not be possible without methodological adaptations from complexity and systems theory.

#### 2.2.5. Ampicillin/Sulbactam

Ampicillin was applied at 581 patient days, whereas ampicillin/sulbactam was given at 2927 patient days. This is very probably a consequence of the high resistance rate of many pathogens to ampicillin, as can be seen from Table 27, which contains the number of antibiograms stratified according to the sensitivity class determined. Ampicillin was introduced in 1961 (patent 1962, cf. PubChem) and is now also an oldie on the market. A literature search shows that numerous alternatives have been discussed due to emerging resistance to ampicillin. Comparisons and combined effects, including adverse events, of ampicillin and amoxicillin were discussed particualarly frequently (well over 10,000 hits in PubMed). The fact is that amoxicillin due to its oral formula is hardly used on ICUs, and ampicillin has been almost completely replaced by the combi-drug ampicillin/sulbactam. The combination of amoxicillin/clavulanate, also a *β*-lactam/*β*-lactamase combination, is an alternative to ampicillin/sulbactam but is hardly used on the study hospital ICU. We therefore restrict the following analysis to ampicillin/sulbactam.

Ampicillin/Sulbactam is a combination of penicillin-derived ampicillin and the additive sulbactam, which is an inhibitor of bacterial beta-lactamase like tazobactam. Ampicillin/Sulbactam is effective against numerous Gram-negative as well as Gram-positive bacteria, but it struggles with an increasing resistance rate. The most common antibiograms were created for the pathogens shown in Table 28.

The time course of the proportion of ineffective ampicillin/sulbactam consumption is shown in the left panel in the third row of Figure 3 with estimated mean and standard deviation of 0.34 and 0.09, respectively. A moderate secular increase can be observed (red line derived from a simple linear model fit). As for the previously discussed antibiotics, an extended linear regression with total ampicillin/sulbactam consumption as an additional covariate reveals a relatively strong positive impact (*β* = 0.028) on the proportion of ineffective treatments (cf. Table 29). Moreover, a moderate secular increase can be confirmed (*β* = 0.009). However, the main driver of developing resistance is the amount of total consumption rather than an independent time effect.

As for the antibiotics discussed above, we again produce a summary table of the epidemiological parameters stratified by observed susceptibility (Table 30) and are surprised to find that the mortality rate for the resistance stratum is significantly lower than for the susceptible class. This seemingly paradoxical result of a protective effect of resistance not only suggests the existence of unknown confounders but also blatantly demonstrates the enormous complexity of a situation characterized by antagonistic interactions. It is worth noting that even the ICU duration is slightly shorter for the resistance class. In addition, length of hospitalization is only moderately prolonged. The assumption that resistance can be reduced with simple scheduled cycling is now finally being reduced to absurdity. This result points to a possibly only implicit logic of the clinical decision to perform the choice of the antibiotic dependent on the vulnerability of the patient. A further, perhaps somewhat bold assumption is that ampicillin-resistant bacteria are less harmful than susceptible bacteria, whereby the therapy takes effect too late in the latter.

#### 2.2.6. Linezolid

Linezolid is a relatively new antibiotic (patent 1999, cf. PubChem) out of the class of oxazolidinones and is classified as a reserve antibiotic, i.e., a drug of last resort. In ICUs with critically ill patients, it is used quite frequently, even in approx. 10% of applications with existing resistance to this antibiotic. Refer to Table 31 for a sensitivity-stratified summary of patient characteristics. The substance is an alternative to vancocymin. Therefore, by far the most frequent use of linezolid was to combat *Staphylococcus* sp. (methicillin-resistant *Staphylococcus aureus* and coagulase-negative staphylococci) and *Enterococcus* sp. infections, with *Staphylococcus epidermidis* and *Enterococcus faecium* being by far the two most frequent addressees.

The time course of the share of inefficient treatment due to resistance is shown in the right panel of the third row in Figure 3. It fluctuates around the average of 10% with a standard deviation of 10% as well, in line with the observation that the proportion of inefficient treatment occasionally drops to zero within intervals. The share of inefficiency more or less follows total consumption; however, consumption never drops below 0.25*DD*. This gives rise to the speculation that resistance can be avoided if total consumption remains below a certain threshold.

A linear regression with time and the amount of total consumption as independent variables once more confirm that the overall consumption is the main driving force for the share of resistance (see Table 32).

#### 2.2.7. Ciprofloxacin and Other Fluoroquinolones

Fluoroquinolones like ciprofloxacin (patent 1985, cf. PubChem) carry a high risk of severe side effects, which are presumably associated with the severely debilitating multi-systemic syndrome ME/CFS (cf. [35]). This is an excellent reason to mention this substance class here with special emphasis. Despite the existing warning “Red Hand Letter” [36], some substances from the class of fluoroquinolones are still approved in Germany and are administered relatively frequently. Fluoroquinolones have a broad spectrum of activity against most Gram-negative and Gram-positive bacteria but should only be used, if at all, for severe infections, particularly in critically ill patients. The Committee for Medicinal Products for Human Use (CHMP) of the European Medicines Agency (EMA), and subsequently the BfArM (which is the German drug regulatory authority), essentially restricted the indication for levofloxacin to community-acquired pneumonia and classified it as a reserve antibiotic in May 2012. In the German ABS S3-guideline [16], fluoroquinolones are considered to be readily available, but the side effect profile and the red hand letter are not mentioned in any sentence.

According to BfArM [36], the chinolones cinoxacin, flumequin, nalidixic acid, and pipemidic acid have never been registered in Germany or have not been registered for a long time. In Germany, registered fluoroquinolones are ofloxacin and norfloxacin, as well as those also prescribed at the studied ICU, which are ciprofloxacin (i.v., n = 2384), levofloxacin (i.v., n = 138; p.o., n = 13), and moxifloxacin (i.v., n = 1035). The time course of the proportion of consumed ciprofloxacin associated with resistance is shown in Figure 3, bottom left panel. Mean and standard deviation are 0.23 and 0.13, respectively. Once more, a clear annual periodicity can be observed. Moreover, the proportion of ineffective consumption apparently follows a strongly decreasing secular drift.

In this case, however, a clear downward trend in overall consumption can also be observed (Figure 6). The confidence band of the regression line shown in the plot is almost unrecognizably narrow, and the regression coefficient is estimated to be −0.19 units (corresponding to DDD) per year. This trend can presumably be explained by the increasing consideration of the warning issued in the red hand letter, in particular due to the almost abrupt decline observed from 2019 onwards (roughly *t*~1000). The observed decline in total consumption proves to be a possible confounder in the context of an extended regression (see Table 33); in any case, total consumption as an additional covariate contributes to a massive correction of the secular trend from the simple regression. It seems plausible that the temporal decrease in the proportion of resistance is actually a consequence of the massive decrease in the total quantity consumed.

In sharp contrast to the time courses of piperacillin and meropenem doses, the average dose per application of ciprofloxacin decreases over time in form of an exponential decay. The corresponding time course is shown in the first row, right panel, of Figure 5 using a logarithmized time scale in order to stress the almost perfect exponential decay by means of a linear regression line fitted to the logarithmized data. The according regression coefficient turns out to be −78.81. The recommendations to exercise restraint when administering fluoroquinolones obviously also had an effect on the dose administered per application. However, it remains an open question whether there is a trade-off between the reduction of resistance and the reduction of adverse events.

To clarify in which cases fluoroquinolones were prescribed in the analyzed clinical dataset, we reduce the laboratory dataset to the corresponding treated patient IDs. Table 34 lists the 10 most frequently isolated germs among this patient group. It can be seen that the infection types are essentially similar to those found in patients treated with piperacillin/tazobactam or meropenem. The frequencies of observed fluoroquinolone sensibilities are listed in Table 35. Of note, Table 34 and Table 35 refer to all three fluoroquinolones administered at the studied ICU.

Finally, the summary Table 36 of epidemiological parameters including death, stratified by the observed sensitivity class, reveals a slight non-significant increase in mortality due to resistance. The significantly smaller *p*-value of 0.08 with regard to an association between sex and sensitivity class is at least worthy of note, even without falling below the significance threshold. Noteworthy is the fact that fatigue syndromes, one of the most worrying adverse effects of fluoroquinolones, are more prevalent in women. Further studies are required to provide clarity. Irrespective of this, it should be made clear once again at this point that this is a small exploratory study that cannot generate more than hypotheses anyway. However, it should also be clear that a possible association between sex and sensitivity further complicates the development of a good ABS.

#### 2.2.8. Cefuroxim

Cefuroxim (patent 1976, cf. PubChem) shares with vancomycin the counterintuitive behavior of showing a decreasing proportion of ineffectiveness with increasing overall consumption (see Table 37 for the corresponding esimates from linear regression). In this case, we explicitly present the regression results including the coefficient for the interaction term. The increase in resistance fraction in response to the covariation of time and total consumption follows an intuitive understanding, but the negative independent effect of consumption is somewhat perplexing. An inspection of the time course of the administered dose per patient, shown in Figure 5, suggests an explanation for this peculiar behavior, as the dose shows a clearly increasing trend.

#### 2.2.9. Correlation of Discharge Destination and Sensitivity Class with Regard to All 8 Main Antimicrobial Representatives

As shown in Table 9, a simple linear regression of the proportion of inefficient treatment on mortality yields a negligible effect. We now examine the phenomenon from a different perspective. In the population of patients treated with any of the eight main antibiotic agents, the summary Table 38 of patient characteristics stratified by sensitivity classes reveals a disturbing effect: a lower mortality in the subgroup that was treated inefficiently is observed. However, if the ampicillin-treated patients are excluded, there is no sensitivity-specific difference in mortality. The prolonged hospitalization time is the only constant result.

### 2.3. Correlations Between the Antibiotics Consumption Time Series

The mutual correlations of the proportions of the consumption time series associated with resistance may provide initial indications of a sensible cyclical allocation strategy (cf. scatter plots in Figure 7 for a selected set of pairs and Figure 8 for a correlation matrix corresponding to the seven most applied antibiotics). For example, replacing an antibiotic for which resistance is increasingly developing with another antibiotic whose resistance rate correlates negatively with the first substance could be a possible strategy. However, it quickly becomes apparent that the aforementioned systematic annual fluctuations in the consumption shares are generally not simultaneous but are shifted against each other with a time lag. It can therefore not be ruled out that this is a case of cross-resistance [17,18,19], which gives rise to time-shifted correlations between the consumption time series, which we will examine below. This view is justified by the fact that the scatter plots of the time series shown in Figure 7 each apparently represent a trajectory belonging to a nonlinear dynamic [37]. They provide some evidence that a chaotic dynamic is the driving force behind the generation of resistance. We do not elaborate on this aspect here but recommend considering it in-depth in further studies.

### 2.4. Mutual Time Lag Correlations Between Antibiotic Consumption Time Series in the Resistance Regime

Cross-resistance [17,18,19] is probably one of the biggest challenges in the context of ABS. If the acquisition of resistance to a certain antibiotic causes the bacteria to become resistant to a completely different antibiotic that has hardly been used so far, then success in switching to this other antibiotic is thwarted from the outset, possibly without any chance of anticipation.

It is almost impossible to prove the existence or effect of cross-resistance on the basis of a retrospective evaluation of secondary data, but in this section, we will at least try to provide some indications and clues. The underlying speculation is that cross-resistance becomes apparent via cross-correlations of the consumption time series labeled as ineffective, taking into account time shifts. In other words, the increase in resistance to an index antibiotic should be reflected in the increase in resistance to a secondary antibiotic with a time lag. Figure 9 shows such so-called time lag correlations (via mutually shifted time series), namely all pair correlations with the ineffective proportion of the piperacillin consumption density as reference time series paired with time series from the selection of the eight most frequently used antibiotics. Because caspofungin has been skipped here as antimycotics are ignored for the time being and the autocorrelation of piperacillin is not relevant here, six pair correlations with a time lag remain. The rationale here is that the development of resistance to some antibiotics runs ahead of the corresponding development of resistance to piperacillin but lags behind for others, depending on which antibiotic of the respective pair is the driving force.

For each correlation pair, the two corresponding time series are shown per panel in Figure 9, taking into account the time shift that maximizes the cross-correlation, with piperacillin as the reference. The time shift and the resulting maximum correlation coefficient are indicated together with the designation of the analyzed pair within the plots. To show that these are not just ordinary correlations but rather dynamic–mechanistic dependencies that control the effectiveness of the antibiotics involved, we take the example of the piperacillin–vancomycin pair and present the reconstructed trajectory graphically in Figure 10. Phase space reconstructions (occasionally also called attractor reconstructions) are frequently used in applied nonlinear dynamics to infer functional dynamic–kinetic relationships from observed time series, and more recently also in infection biology [37]. To this end, we search for the time lag that leads to the first local minimum of the time lag correlation and use it to reconstruct the trajectory in the optimal affine-transformed phase space. An irregular, possibly chaotic oscillatory dynamic can be recognized quite clearly. All other antibiotic pairs provide similar results. Without overestimating this qualitative result, it at least indicates an irregular cyclic variation between the ineffective fractions of antibiotic consumption.

## 3. Discussion

The central message of this work is twofold, namely that firstly, the integration of clinical expertise, especially in the field of intensive medical care with the expertise from the microbiology laboratory, but also with the expertise of the mathematical and statistical modeler, represents a tremendous challenge; secondly, there is no time to lose in tackling this integration. Although we limited our descriptive or semi-quantitative analysis to a few problem areas in the field of the development of resistance of microbiological organisms to antimicrobial treatment, we can still report an overwhelming complexity of the clinical situation regarding nosocomial or imported infections and antimicrobial resistance. Needless to say, due to the nature of any observational study, the evaluation presented is also full of limitations. Although we presented a reasonably comprehensive basket of analyses, the study is far from exhaustive. Many fundamental questions remain unanswered; nevertheless, we are tackling the task of structuring the clinical data and processes required for the structured optimization of ABS.

We would like to convey the insight that overcoming the resistance crisis should not be demanded with a raised index finger against the clinicians, but rather that the undoubtedly existing clinical expertise should not only be recognized but should even be the basis for deriving expert systems within the framework of mathematical support systems. The main direction of impact should be the optimization of actions on the basis of both explicit and implicit clinical knowledge through coordination in a complex environment, in which statistical optimization methods, possibly also AI-based systems, can unfold their advantages. Given the huge number of potential predictors for the development of resistance, which we initially had to ignore in this exploratory work, sophisticated evaluation algorithms from the machine learning pool are almost inevitable. With this in mind, it should have become clear that the interdependence of global and local fields of action and impact, i.e., a synergistic perspective, is crucial.

To be specific, we demonstrated, perhaps surprisingly, that the secular increase in resistance is quite moderate for most antibiotics, whereas the dependence on total consumption is strikingly high. But also here, there are two counterintuitive observations, namely for vancomycin as well as for cefuroxim: we obtain a decreasing proportion of resistance with increasing total consumption. An explanation based on the changing dose per application over time is initially obvious, but the fact that the dose for cefuroxim shows a clearly increasing trend but the dose for vancomycin shows a markedly decreasing trend requires a more precise analysis of the blood and tissue concentrations of the respective substance, i.e., precise drug monitoring.

It would be presumptuous to draw far-reaching conclusions from this, but the aforementioned observations make us confident that it is indeed feasible to curb resistance both through intelligent variations in consumption and adaptations in dose per application. This ambitious project is, of course, made more difficult by cross-resistance, which is indicated by the results of time lag cross-correlations. Thus, increasing the overall consumption of antibiotics in the population increases the risk of resistance development, but what about low overall consumption but frequent intra-individual repeated use, i.e., an overuse on the individual level? Our provisionally derived results point to a positive correlation between the number of consecutive AB treatments within patients and a share of inefficiency of treatment due to resistance.

Less surprising than secular shift and dependence of total consumption is the observation that mortality is higher among AB-treated patients, where we assume “AB treated” as a surrogate for “infected”. In addition, length of ICU stay and, more generally, duration of hospitalization turned out to be significantly prolonged within the AB-treated cohort. A fatal effect of AB treatment itself cannot be ruled out but is beyond the scope of what can be investigated here.

The assumption that mortality would be even higher without AB treatment seems plausible but can at best be answered indirectly retrospectively on the basis of the available dataset. For an indirect conclusion, for example, a presumably ineffective or at least less effective treatment due to resistance could be interpreted as a control in comparison with susceptible cases. For this purpose, we analyzed the epidemiological characteristics of the patients within the AB treatment stratum, including deaths and duration of hospitalization, stratified by assigned sensitivity class using summary tables. The observed effect sizes differ among the eight common antibiotics studied individually, but they all have in common at least a moderate increase in mortality with ineffective treatment, i.e., the presence of resistance, as well as a significant increase in the duration of hospitalization. Once again, there is a somewhat puzzling exception, namely the cohort of patients treated with ampicillin. The resistance rate for ampicillin is huge anyway, but the increase in deaths in the cohort under treatment with given susceptibility of the pathogen is somewhat astonishing to say the least. The assumption that susceptible germs could be more harmful and that the therapy is too slow to respond is, of course, just one of the possible speculations that must now be urgently reviewed. Or are there other or further synergistic or antagonistic interactions between pathogen variants and host, such as a disruption of the microbiome in the case of “successful” treatment of susceptible bacterial variants? It is clear that future research must be open in all directions.

In the population of patients treated with any of the eight main antibiotic agents, the same picture emerges, namely a lower mortality in the subgroup that was treated inefficiently. However, if the ampicillin-treated patients are excluded, there is no sensitivity-specific difference in mortality. The prolonged hospitalization time is the only constant result. It does not need to be emphasized that the length of stay is in a conditional relationship to the death event, but modeling competitive outcomes is far beyond feasibility here.

In summary, a number of hypotheses can be generated from the analyses carried out that still need to be tested, but unfortunately, only a few concrete conclusions can be drawn. The obvious conclusion is that as long as precise knowledge of the dynamics is lacking, an equal distribution of the total consumption of the individual antibiotics should be aimed for, as the concentration of consumption on a small number of antibiotics increases the proportion of resistance. This is mainly due to the fact that hardly any antibiotic has a significantly lower proportion of ineffectiveness.

However, a closer look at the dynamics reveals that there are cyclical variations in resistance shares with approximately annual periods, which ultimately actually suggest a cyclical allocation strategy adapted to these cycles. Furthermore, we found evidence that these cycles associated with each antibiotic are systematically shifted in pairs, possibly indicating retarded cross-correlations. It is obvious that these transfer processes obey nonlinear dynamics.

That said, we draw attention to our observation that both multiple infections, often enough even with all known pathogens classified as particularly dangerous, and multiple antimicrobial therapies are the norm rather than the exception. Far-reaching interactions, which of course also include cross-relationships, can hardly be underestimated. Simply switching from one mainly used antibiotic to another main application does not really help here.

It need not be emphasized that any ABS strategy is limited by clinical indications that per se restrict the use to a few antibiotics. However, it must be warned against preferential allocation that is not based on well-founded evidence, which could be the case with piperacillin, for example. Let us then summarize our observations and obvious conclusions regarding the use of fluoroquinolones. The fact is that in the ICU studied, the use of these substances, which is strongly warned against in a red hand letter, was reduced to almost zero over the course of the observation period. Resistance to these substances also strongly suggests that they should not be reintroduced.

## 4. Materials and Methods

### 4.1. Study Type

The study conducted is a retrospective observational study. With regard to ABS, the study is clearly exploratory in nature. Causal implications are not feasible within the framework of such a study. It is intended to generate hypotheses on trajectories of the genesis of resistance and the effects of resistance from the recorded hospital health data, ultimately performed in order to design potential monitoring strategies and decision support.

The study was conducted using microbiological and medical data from patients treated at the Center for Intensive Care Medicine and Intermediate Care at the Knappschaft Kliniken University Hospital in Bochum, Germany. Data from one of the intensive care units located in the center, where mainly surgical-operative cases from the disciplines of general and visceral surgery, transplant surgery, neurosurgery, trauma surgery, and oral and maxillofacial surgery are treated, were included. To a lesser extent, patients from the specialist disciplines of internal medicine, hemato-oncology, and neurology were also treated in the ward. The ward has a total of 21 high-care treatment places, which are divided into 13 rooms (5 single-bed rooms and 8 double-bed rooms). These rooms are located next to each other on a long corridor.

### 4.2. Primary Clinical Systems Database

After obtaining an ethics vote for our study “Model-based optimization of antibiotic stewardship strategies (Opti4ABS)” from the Ethics Committee of the Medical Faculty of the Ruhr University Bochum (register number: 23-7836-BR), the clinical data were retrieved. The evaluation included patients who were treated in the interdisciplinary surgical intensive care unit of the Knappschaft Kliniken University Hospital Bochum between 1 January 2016 and 31 March 2023. A total of 7718 patients with 9201 ward stays were identified, i.e., some patients had multiple hospitalizations. The analyzed ICU is a surgical intensive care unit that does not generally include the care of pediatric patients, but in very rare cases, emergency care may be provided for newborns or children who remain in the ICU until stable transport is possible. However, these patients do not play a role in the analysis presented here.

The SQL-based data query was then performed from the database of the electronic health records (hospital information system and patient data management system). The derived data were made available to our project in csv format and included the following entities:Demographic data including principal diagnosis;Secondary diagnoses and procedures;Location information (rooms where the patient stayed in the ICU);Documented antibiotic administration;Laboratory values/measurements/scores.These data were then checked for plausibility, corrected if necessary, and converted into an initial structured format.

The dataset was then extensively anonymized with the aim of retaining the granularity of the data required for the analysis, but at the same time ensuring that no reference to natural persons could be performed at the end. The anonymization therefore included the following aspects, among others:
Calculation of the age at hospital admission and removal of the date of birth;Removal of the patient-identifying case and patient number;Removal of other personal details (if available in the data record), e.g., surname, first name, address, other identification numbers, etc.;Conversion of the date reference of the individual data records into a treatment day specification from which it is no longer possible to draw conclusions about the date.

For the project, it is necessary to map the spatial and temporal relationship of patients treated on the ward at the same time (e.g., to map the transmission of resistance between patients). In order to anonymize the dataset, all dates for each patient were converted into a uniform global treatment day (as a numerical number: 0, 1, 2, … 989, 990, 991, …, etc.). The uniform reference date (day 0) was only known to the clinical study center at the University Hospital Knappschaftskrankenhaus Bochum and was destroyed after completion of the data preparation and analysis process. The date was explicitly not set to 1 January 2016 (start of inclusion of patients) but to a random date thereafter. Similarly, the end date was not 31 March 2023 (end of patient inclusion) but a random date before this. This date was also not communicated externally and was destroyed at the end.

### 4.3. Antibiogram and Microbiological Data

The clinical dataset was then expanded to include the specific microbiological findings (including resistance determinations). As these data are not primarily held in the hospital information system, it was necessary to program an additional interface to the data-holding system, which automatically extracts the relevant findings and processes them in accordance with standards, as well as anonymizing them. The microbiological data contain information on the pathogens isolated and microbiologically examined, in particular the organ/tissue sampled and the quantity, as well as their susceptibility to antibiotics. In order to create the antibiograms, almost every isolated pathogen of every patient was subjected to multiple antibiotic provocations, and the corresponding sensitivity was determined by applying the threshold values to the MIC. Only a few pathogens were not challenged.

At the turn of the year from 2018 to 2019, the previously so-called intermediate sensitivity category *I* was redefined by EUCAST as “susceptible, increased exposure” [7]. The definition also changed the application recommendation. Antibiotics classified as “intermediate” were generally not prescribed before 2019 but were prescribed afterwards, albeit at an increased dose. As the observation period of the data examined here begins in 2016, this change in definition must be kept in mind.

### 4.4. Statistical Analysis and Mathematical Modeling

#### 4.4.1. Descriptive Statistics

Descriptive statistics are presented both graphically as well as either through frequency or contingency tables or through correlation (including time lag cross-correlation) and regression analyses, respectively. The usual 5% significance level is used for comparisons and regressions as well as 95% confidence intervals, when appropriate. The statistical tests used in each case are noted in the footer of the corresponding tables and include chi-squared or Fisher’s exact tests for categorical variables, t-tests or, if applicable, rank tests for continuous variables. All analyses and graphs were created using the statistical programming language R [38] version 4.3.3.

#### 4.4.2. Consumption Density Time Series of Antibiotics in the Presence of Resistance

The dataset contains the daily doses of antibiotics administered per patient, expressed in the specific unit. The normalization of these daily doses by dividing them by the median consumption value taken over the entire observation period and across all patients results in a standardized value within the clinic, which is analogous to the general calculation of daily defined doses (DDDs) but arguably closer to the often preferred recommended daily doses (RDDs). Different antibiotics are therefore comparable within the clinic in terms of daily consumption, and because no interclinical comparisons are made, the type of standardization has no effect.

Moving average calculations over intervals of 180 days are applied to the daily consumptions of eight of the most commonly prescribed antibiotics, stratified by antibiogram results. In other words, for each antibiotic *A* and each of the four strata s∈{S,I,R,NA}, the normalized and smoothed quasi-continuous consumption $A_{s}\left(t\right)$ as a function of time is calculated along with the proportion
(1)$$p_{A_{R}}\left(t\right)=\frac{A_{R}\left(t\right)}{A_{R}\left(t\right)+A_{I}\left(t\right)+A_{S}\left(t\right)+A_{NA}\left(t\right)}$$
of the resistant stratum, i.e., the proportion of ineffective applications. Hereby, the fourth stratum *NA* refers to an unknown sensitivity either because the antibiogram yielded an ambiguous result or because no antibiogram was created. Therefore, the fraction $p_{A_{R}}\left(t\right)$ has to be interpreted as the lower bound of the real proportion. Arguably, the smoothing process via the moving average, as well as the previously carried out standardization of the consumption quantity to the overall median consumption of the given antibiotic, justifies calling $A_{s}\left(t\right)$ the consumption density. Both consumption densities $A_{s}\left(t\right)$ and the share of inefficient treatment $p_{A_{R}}\left(t\right)$ are conceived as outcome variables in correlation and regression analyses, but without any claim to causal explainability. The rationale behind this approach follows from an approach used in compartmental modeling of infection epidemiological processes. This approach makes it possible to use the repertoire of methods from the field of mathematical dynamic modeling. This includes the calculation of mutual correlations and delay correlations, as well as phase space representations (cf. [25,37]).

Occasionally, more than one pathogen was present in a patient’s sample, and also in some cases, more than one antibiogram was carried out with respect to a specific antibotic-pathogen pair. In these cases, a specific antibiotic *A* was labeled *R* (i.e., ineffective) if at least one antibiotic–pathogen pair with respect to all tested pathogens yielded resistance. This scheme was applied hierarchically to the series *R*, *I*, *S*, thus, a specific antibiotic *A* was labeled *S* if and only if all tested *A* − *pathogen* pairs yielded susceptible pathogens.

Fast Fourier transform-based spectral analyses are carried out for selected time series that are considered particularly relevant in order to investigate seasonal fluctuations and other systematic cycles. The results are presented graphically in the form of spectral densities. Furthermore, time lag cross-correlation functions are computed, and the shifted time series are presented graphically.

## 5. Outlook

The scope, complexity, and multi-layered nature of the data to be analyzed and the dynamics to be explained required us to focus on only some of the aspects of interest. The temporal variations of the resistance proportions, as well as the dependencies on the total consumption, but also in particular the time lag correlations of the consumption time series under consideration, testify to a heterogeneity both in the substance consumption and in the infection incidence, which we have not yet sufficiently considered. In addition, the room numbers of the patients also provide spatial information that can, at least in principle, be used to describe the transmission dynamics. We will dedicate a subsequent publication to the analysis of spatiotemporal heterogeneity and transmission dynamics. Entropy and diversity measures will then play a decisive role. Preliminary work on this has already been published [25]. This and the planned publication will then serve as a basis for the design of a monitoring and decision support system.

To describe our strategy for taking on the enormous challenge of a promising ABS, we use the motto from nonlinear dynamics: harnessing chaos. Just as a rider receives the best out of their horse by harmonizing the target dynamics with its own dynamics, for a successful ABS—and that is, an informed ABS—we must first understand the intrinsic modes of the complex dynamics behind it. We hope that we have been able to pave the rocky road to harnessing antibiotic resistance a little.

## Figures and Tables

**Figure 1 antibiotics-14-00561-f001:**
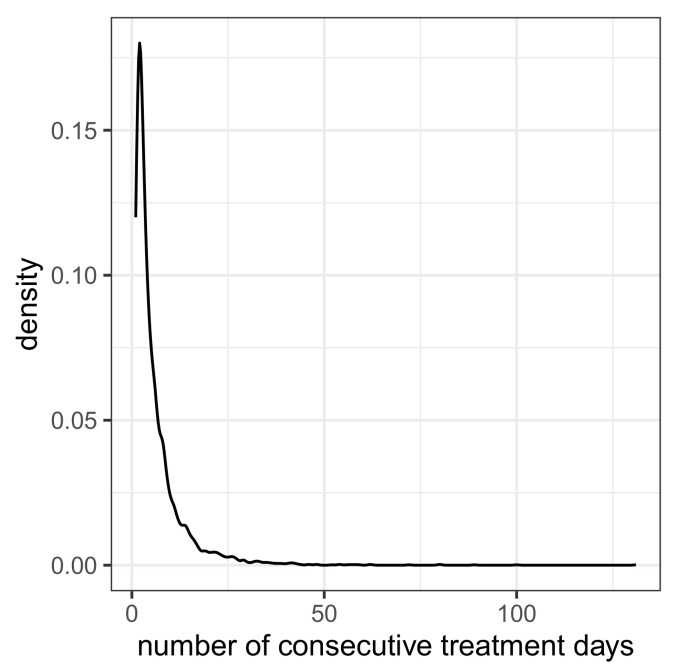
Density curve for the duration of treatment sequences (i.e., number of consecutive treatment days).

**Figure 2 antibiotics-14-00561-f002:**
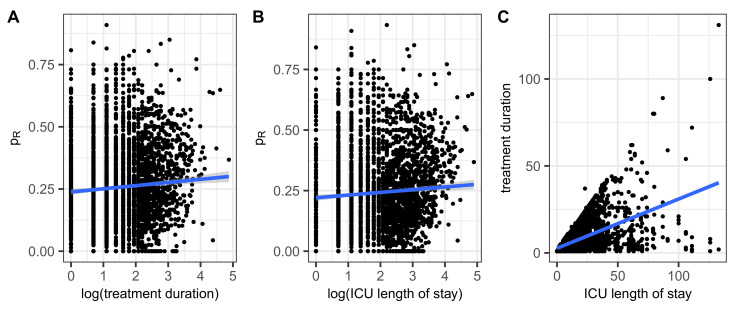
Share of resistance within an antibiogram belonging to a patient/treatment sequence versus (**A**) the length of the treatment sequence and (**B**) the length of the ICU stay. Panel (**C**) shows the scatter plot of the two durations to check consistency. Linear regression lines added in blue.

**Figure 3 antibiotics-14-00561-f003:**
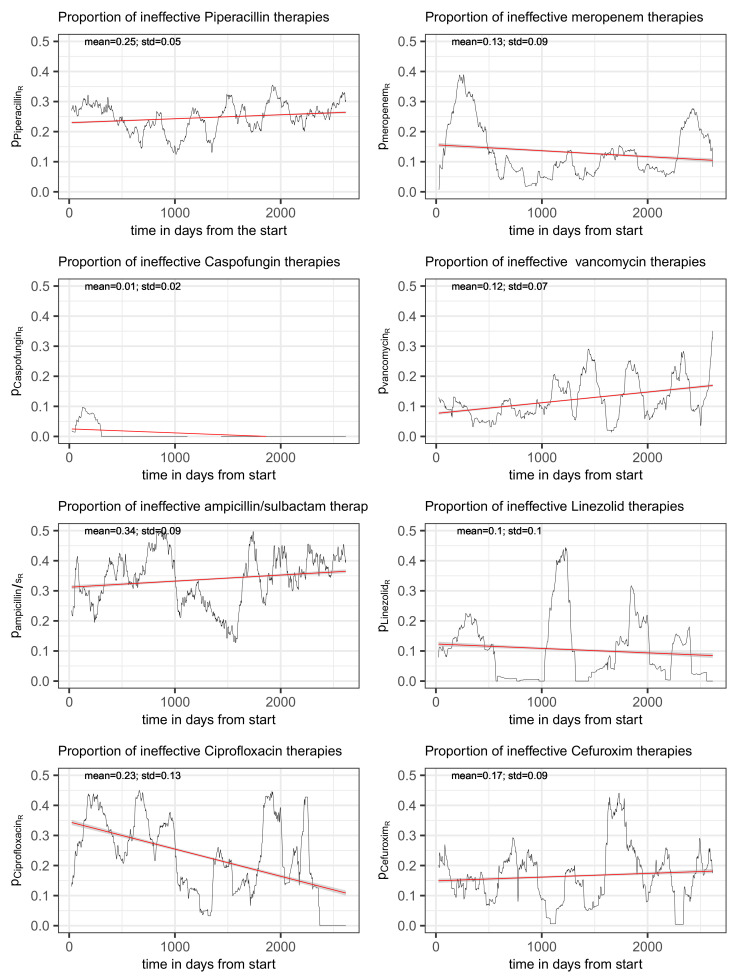
Time series of the proportion of the total consumption of an antibiotic to which resistance of at least one of the treated pathogens was detected, shown for the eight most frequently administered antimicrobials. The red line represents a simple fitted linear model to visualize the trend.

**Figure 4 antibiotics-14-00561-f004:**
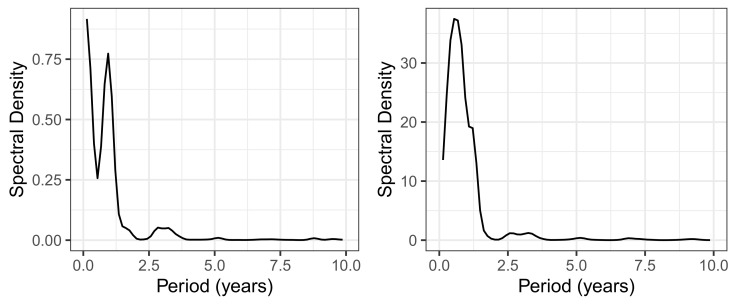
Spectral density for the piperacillin consumption density time series. (**Left**): Spectral density of $p_{piperacillin_{R}}\left(t\right)$, i.e., the proportion of ineffective applications. (**Right**): Spectral density of the piperacillin total consumption time series.

**Figure 5 antibiotics-14-00561-f005:**
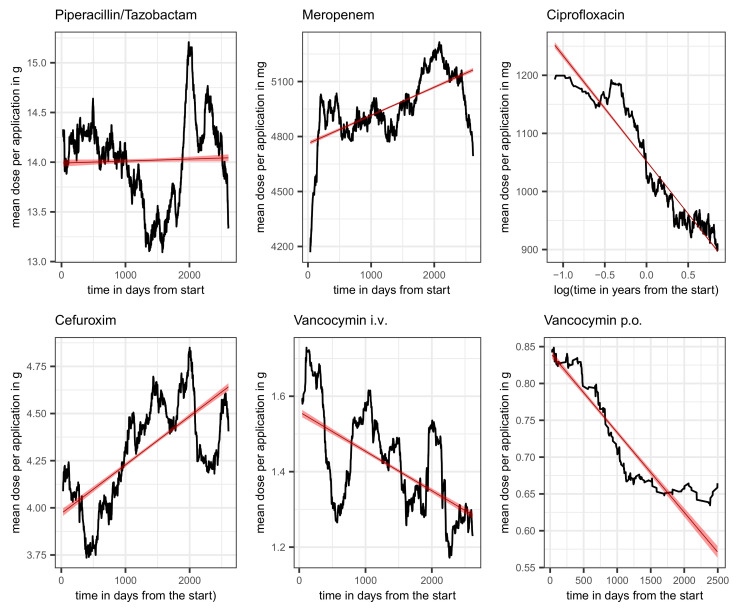
Average dose per application over the course of time (moving average over intervals of 180 days) along with a linear regression line (red) plus confidence band for piperacillin/tazobactam, meropenem, ciprofloxacin, cefuroxim, and vancocymin i.v.; and vancocymin p.o., as indicated in the header line of each panel. The time scale for ciprofloxacin is logarithmized to address the apparent exponential decay.

**Figure 6 antibiotics-14-00561-f006:**
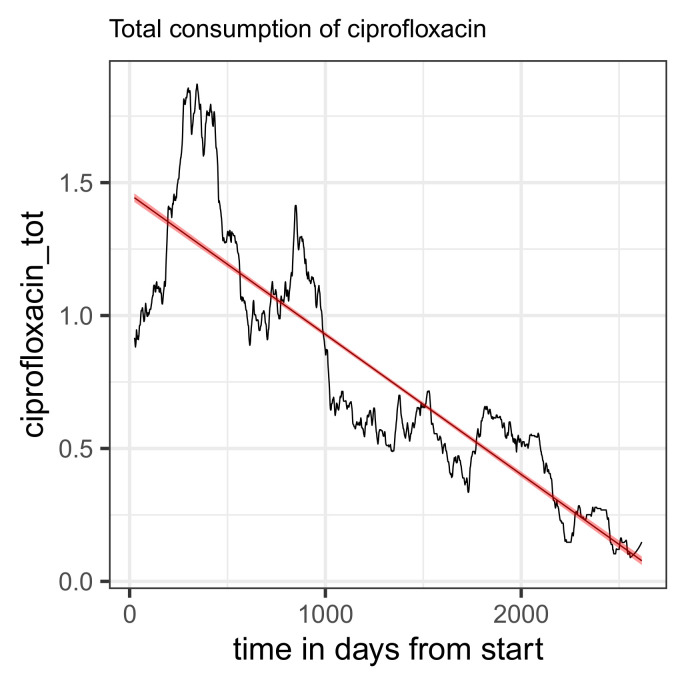
Time series of total consumption of ciprofloxacin (black curve) along with a linear regression line (red).

**Figure 7 antibiotics-14-00561-f007:**
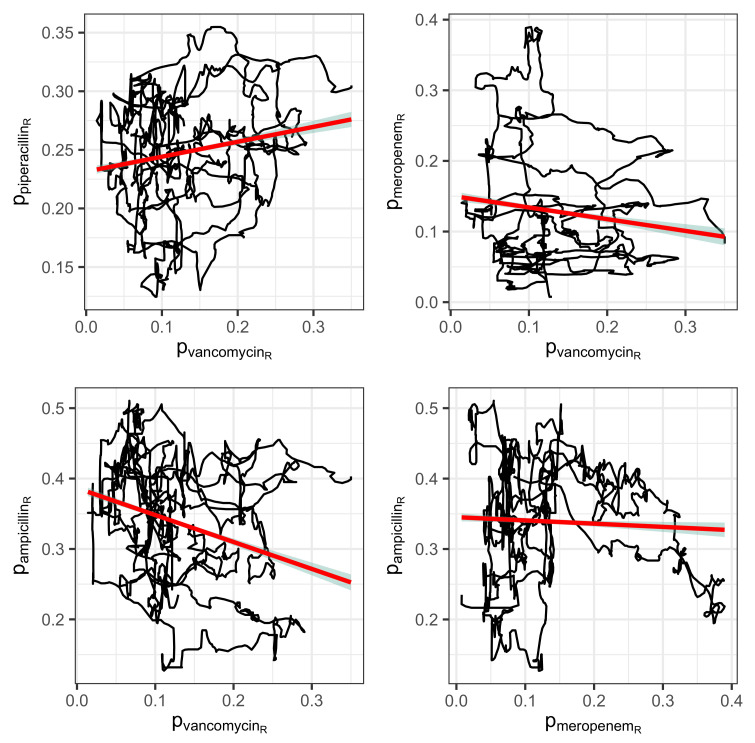
Correlations visualized by means of linear regression lines (red) between consumption time series depicted as trajectory $p_{A_{R}}\left(t\right)$ versus $p_{B_{R}}\left(t\right)$ with respect to antibiotics A and B, respectively.

**Figure 8 antibiotics-14-00561-f008:**
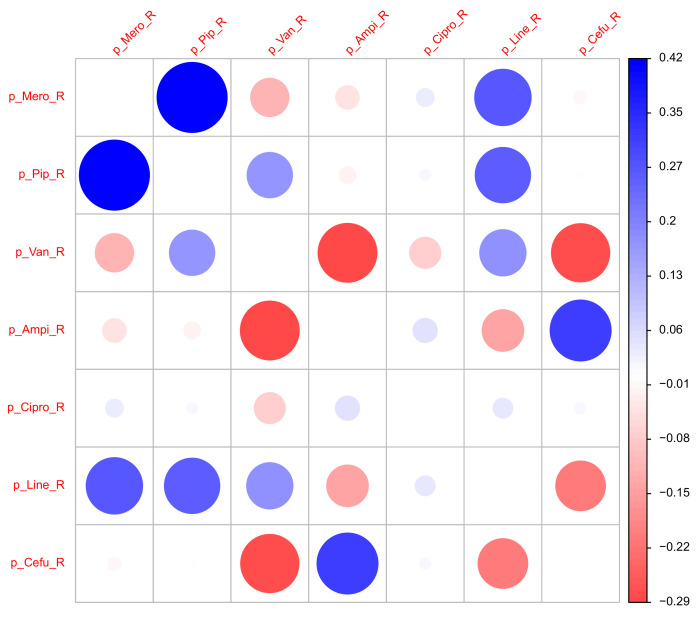
Correlation matrix of time series of the proportions of ineffective consumption.

**Figure 9 antibiotics-14-00561-f009:**
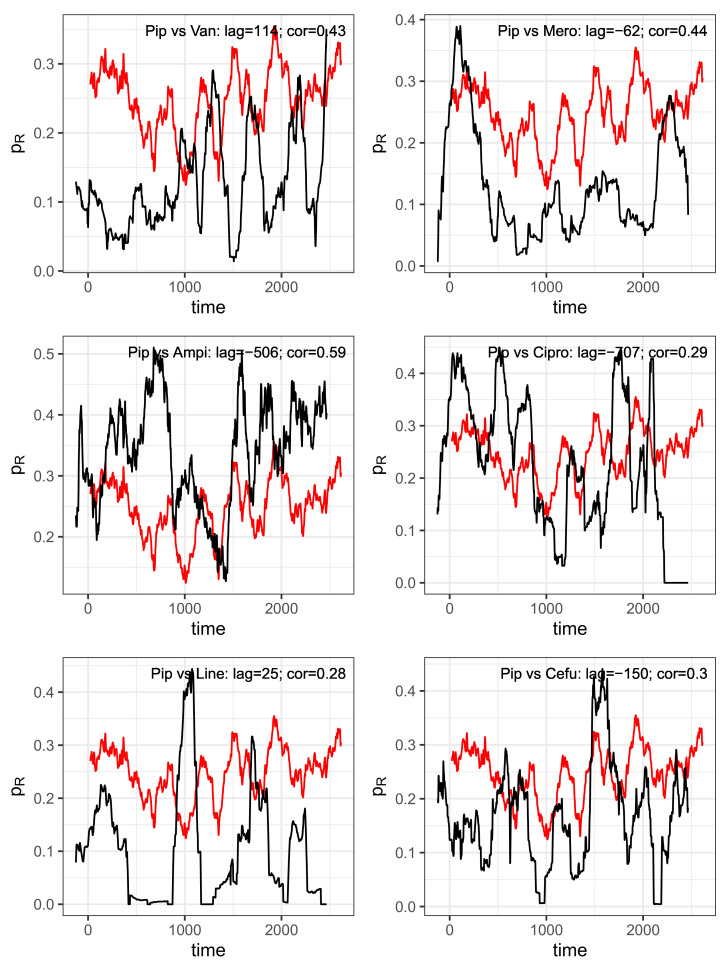
Pairs of time series $p_{Antibiotic_{R}}\left(t\right)$ shifted against each other with the delay derived from maximum cross-correlations. Reference: Piperacillin (“Pip”, red curve). Abbreviations of the antibiotics of the shifted comparison curves (black): Van = vancomycin, Mero = meropenem, Ampi = ampicillin, Cipro = ciprofloxacin, Line = linezolid, Cefu = cefuroxim.

**Figure 10 antibiotics-14-00561-f010:**
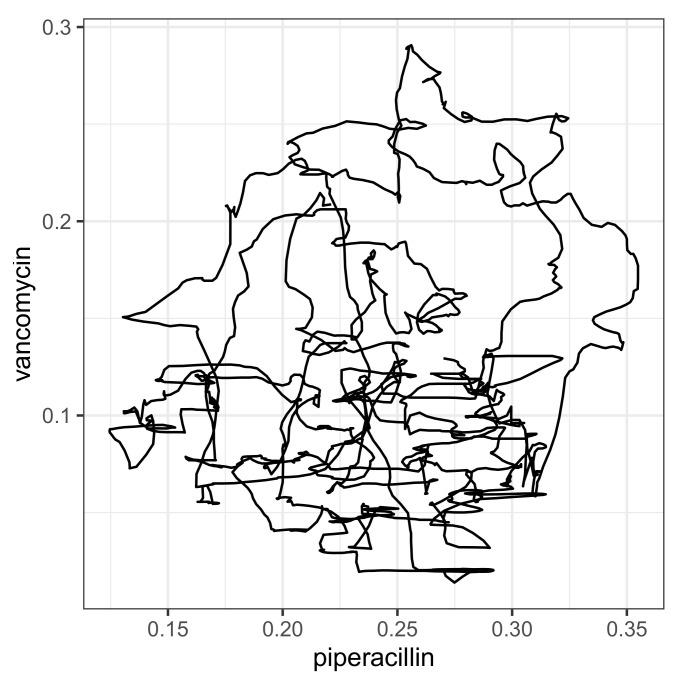
Trajectory produced on the basis of mutually shifted time series $p_{piperacillin_{R}}\left(t\right)$ and $p_{vancomycin_{R}}\left(t-dt\right)$.

**Table 1 antibiotics-14-00561-t001:** Summary table of demographic/epidemiological characteristics stratified by sex. Age in years and Age_group are dichotomised in two groups at the median. Subjects are ward stays. Three ward stays are missing due to missing sex indication.

Characteristic	Female N = 4141	Male N = 5057	*p*-Value
Age (Years)	69 (56, 80)	64 (52, 75)	<0.001
Age_group			<0.001
>median age	2271 (55%)	2279 (45%)	
≤median age	1870 (45%)	2778 (55%)	
Hospitalization Duration (Days)	13 (7, 25)	15 (7, 28)	0.002
ICU Duration (Days)	2 (1, 5)	2 (1, 5)	0.004
Discharge Destination			0.002
death	742 (18%)	801 (16%)	
external hospital	1434 (35%)	1686 (33%)	
home	1965 (47%)	2570 (51%)	

Median (Q1, Q3); n (%); Wilcoxon rank sum test; Pearson’s chi-squared test.

**Table 2 antibiotics-14-00561-t002:** Summary table of demographic characteristics stratified by antibiotic treatment flag. Subjects are ward stays.

Characteristic	Overall N = 9201	w/o AB Treatment N = 4281	with AB Treatment N = 4920	*p*-Value
Sex				<0.001
female	4141 (45%)	2120 (50%)	2021 (41%)	
male	5057 (55%)	2161 (50%)	2896 (59%)	
unknown	3	0	3	
Age (Years)	66 (54, 78)	66 (52, 78)	66 (55, 77)	0.4
Destination				<0.001
death	1545 (17%)	487 (11%)	1058 (22%)	
external hospital	3121 (34%)	1405 (33%)	1716 (35%)	
home	4535 (49%)	2389 (56%)	2146 (44%)	
Hospital Duration (Days)	14 (7, 26)	9 (4, 16)	21 (11, 37)	<0.001
ICU Duration (Days)	2 (1, 5)	1 (1, 2)	4 (1, 9)	<0.001

n (%); Median (Q1, Q3); Pearson’s chi-squared test; Wilcoxon rank sum test.

**Table 3 antibiotics-14-00561-t003:** Frequencies of the different numbers of sequences per patient.

**Sequences Per Patient**	1	2	3	4	5	6	7	9	10
**Frequency**	3222	506	123	29	12	4	2	1	1

**Table 4 antibiotics-14-00561-t004:** Frequency table for the occurrences of simultaneous antimicrobial treatments per patient per day.

n_AB Per Patient Per Day	Occurrences
1	14,419
2	7998
3	4169
4	1814
5	701
6	229
7	61
8	21
10	1

**Table 5 antibiotics-14-00561-t005:** Frequency table showing the number of occurrences of each sensitivity class S, I, R, and n.a., respectively.

Sensitivity	Challenges	Distinct IDs
n.a.	537	148
I	12,309	2477
R	41,525	2951
S	101,223	3406

**Table 6 antibiotics-14-00561-t006:** Frequently isolated pathogens.

Pathogen	n
*Escherichia coli*	3091
*Staphylococcus epidermidis*	2664
*Candida albicans*	2299
*Enterococcus faecium*	1745
*Staphylococcus aureus*	1579
*Klebsiella pneumoniae*	1336
*Candida glabrata*	1265
*Pseudomonas aeruginosa*	1126
*Enterococcus faecalis*	644
*Enterobacter cloacae*	614

**Table 7 antibiotics-14-00561-t007:** Results from linear regression: share of resistance versus logarithm of duration of treatment sequence and number of administered antibiotics per day (*n_ABIO_*).

Characteristic	Beta	95% CI	*p*-Value
log(duration)	0.01	0.01, 0.02	<0.001
*n_ABIO_*	0.02	0.01, 0.03	<0.001

Abbreviation: CI = Confidence interval.

**Table 8 antibiotics-14-00561-t008:** Results from linear regression: share of resistance versus logarithm of ICU length of stay.

Characteristic	Beta	95% CI	*p*-Value
log(ICU.Stay + 1)	0.01	0.00, 0.02	<0.001

CI = Confidence interval.

**Table 9 antibiotics-14-00561-t009:** Results from linear regression: share of resistance versus hospital discharge destination.

Characteristic	Beta	95% CI	*p*-Value
Destination:			
external hospital	-	-	
home	−0.001	−0.015, 0.012	0.865
death	0.002	−0.014, 0.018	0.789

CI = Confidence interval.

**Table 10 antibiotics-14-00561-t010:** Results from linear regression: share of resistance versus number of distinct treatment sequences per patient (n_seq).

Characteristic	Beta	95% CI	*p*-Value
n_seq	0.04	0.03, 0.05	<0.001

CI = Confidence interval.

**Table 11 antibiotics-14-00561-t011:** The 9 most frequently administered antibiotics.

Antibiotic	n_Days	n_Indiv
Piperacillin/Tazobactam i.v.	8227	1752
Meropenem i.v.	6643	1011
Caspofungin i.v.	3598	439
Vancomycin i.v.	3455	637
Ampicillin/Sulbactam i.v.	2927	903
Flucloxacillin i.v.	2448	406
Linezolid i.v.	2434	371
Ciprofloxacin i.v.	2384	547
Cefuroxim i.v.	2265	596

**Table 12 antibiotics-14-00561-t012:** Linear regression of the proportion of ineffective piperacillin/tazobactam consumption on time and total consumption. “Mibi Time” refers to the time of the microbiological examination in years from the start of the study period and “Total Consumption” refers to an estimated DDD.

Characteristic	Beta	95% CI	*p*-Value
(Intercept)	0.182	0.170, 0.195	0.000
Mibi Time	0.002	0.001, 0.003	0.000
Total Consumption	0.023	0.017, 0.028	0.000

CI = Confidence interval.

**Table 13 antibiotics-14-00561-t013:** Summary table of performed piperacillin tests stratified by observed sensitivity class.

Antibiotic	Sensitivity	n
Piperacillin	I	352
Piperacillin	R	3242
Piperacillin	S	1459
Piperacillin/Tazobactam		3
Piperacillin/Tazobactam	I	1169
Piperacillin/Tazobactam	R	1313
Piperacillin/Tazobactam	S	3925

**Table 14 antibiotics-14-00561-t014:** Summary table of combinations of performed piperacillin tests.

Antibiotic	n
Both	5017
Piperacillin	36
Piperacillin/Tazobactam	1393

**Table 15 antibiotics-14-00561-t015:** Most common pathogens tested for piperacillin (with and w/o tazobactam) resistance stratified by sensitivity.

Pathogen	Sensitivity	n
*Escherichia coli*	S	2141
*Escherichia coli*	I	650
*Escherichia coli*	R	1736
*Klebsiella pneumoniae*	S	641
*Klebsiella pneumoniae*	R	1213
*Pseudomonas aeruginosa*	S	433
*Pseudomonas aeruginosa*	I	714
*Pseudomonas aeruginosa*	R	813
*Serratia marcescens*	S	576
*Proteus mirabilis*	S	542
*Klebsiella oxytoca*	R	374

**Table 16 antibiotics-14-00561-t016:** Three most common pathogens tested for piperacillin/tazobactam resistance stratified by sensitivity.

Pathogen	Sensitivity	n
*Escherichia coli*	S	1583
*Escherichia coli*	I	650
*Escherichia coli*	R	376
*Klebsiella pneumoniae*	S	641
*Klebsiella pneumoniae*	I	79
*Klebsiella pneumoniae*	R	382
*Klebsiella pneumoniae*		3
*Pseudomonas aeruginosa*	S	217
*Pseudomonas aeruginosa*	I	370
*Pseudomonas aeruginosa*	R	394

**Table 17 antibiotics-14-00561-t017:** Summary table of epidemiological characteristics of piperacillin-treated patients stratified by observed sensitivity.

Characteristic	I N = 185	n.a. N = 747	R N = 309	S N = 511	*p*-Value
Sex					0.7
female	69 (37%)	299 (40%)	130 (42%)	210 (41%)	
male	116 (63%)	448 (60%)	178 (58%)	301 (59%)	
unknown	0	0	1	0	
Age (Years)	70 (60, 80)	68 (56, 78)	67 (58, 76)	69 (59, 79)	0.031
Destination					0.2
death	54 (29%)	216 (29%)	98 (32%)	139 (27%)	
external hospital	82 (44%)	293 (39%)	112 (36%)	227 (44%)	
home	49 (26%)	238 (32%)	99 (32%)	145 (28%)	
ICU Duration	7 (2, 15)	5 (2, 10)	8 (3, 21)	7 (3, 15)	<0.001
Hospital Duration	22 (12, 39)	15 (8, 24)	35 (18, 54)	22 (13, 36)	<0.001

n (%); Median (Q1, Q3); Pearson’s chi-squared test; Kruskal–Wallis rank sum test.

**Table 18 antibiotics-14-00561-t018:** Linear regression of the proportion of ineffective meropenem consumption on time and total consumption. “Mibi Time” refers to the time of the microbiological examination in years from the start of the study period and “Total Consumption” refers to an estimated DDD.

Characteristic	Beta	95% CI	*p*-Value
(Intercept)	0.040	0.030, 0.049	0.000
Mibi Time	−0.024	−0.026, −0.022	0.000
Total Consumption	0.083	0.077, 0.088	0.000

CI = Confidence interval.

**Table 19 antibiotics-14-00561-t019:** Summary table of performed meropenem tests stratified by observed sensitivity class.

Antibiotic	Sensitivity	n
Meropenem	I	222
Meropenem	R	291
Meropenem	S	6685

**Table 20 antibiotics-14-00561-t020:** Most common pathogens tested for meropenem resistance stratified by observed sensitivity.

Pathogen	Sensitivity	n
*Escherichia coli*	S	2617
*Escherichia coli*	R	4
*Klebsiella pneumoniae*	S	1071
*Klebsiella pneumoniae*	I	15
*Klebsiella pneumoniae*	R	17
*Pseudomonas aeruginosa*	S	590
*Pseudomonas aeruginosa*	I	192
*Pseudomonas aeruginosa*	R	214
*Enterobacter cloacae*	S	512
*Enterobacter cloacae*	I	4
*Enterobacter cloacae*	R	5
*Proteus mirabilis*	S	390
*Proteus mirabilis*	R	1
*Klebsiella oxytoca*	S	366
*Klebsiella oxytoca*	I	1

**Table 21 antibiotics-14-00561-t021:** Summary table of epidemiological characteristics of meropenem-treated patients stratified by observed sensitivity.

Characteristic	I N = 32	n.a. N = 307	R N = 86	S N = 586	*p*-Value
Sex					0.8
female	13 (41%)	124 (40%)	31 (36%)	222 (38%)	
male	19 (59%)	183 (60%)	55 (64%)	363 (62%)	
unknown	0	0	0	1	
Age	63 (50, 73)	64 (54, 75)	63 (53, 71)	66 (55, 75)	0.060
Destination					0.12
death	9 (28%)	120 (39%)	35 (41%)	185 (32%)	
external hospital	16 (50%)	101 (33%)	32 (37%)	227 (39%)	
home	7 (22%)	86 (28%)	19 (22%)	174 (30%)	
ICU Duration	9 (3, 29)	7 (2, 14)	14 (5, 27)	10 (3, 20)	<0.001
Hospital Duration	36 (19, 49)	20 (10, 33)	40 (26, 77)	27 (16, 46)	<0.001

n (%); Median (Q1, Q3); Pearson’s chi-squared test; Kruskal–Wallis rank sum test.

**Table 22 antibiotics-14-00561-t022:** Crosstable showing the frequencies of observed sensitivity classes of all pathogens challenged with caspofungin.

Pathogen	n.a.	I	R	S
*Candida albicans*	63	0	0	83
*Candida dubliniensis*	4	0	0	0
*Candida glabrata*	50	0	0	46
*Candida guilliermondii*	1	0	0	0
*Candida krusei*	6	0	0	0
*Candida lusitaniae*	0	0	0	1
*Candida norvegensis*	0	0	0	1
*Candida orthopsilosis*	1	0	0	0
*Candida parapsilosis*	11	1	4	12
*Candida tropicalis*	5	0	0	19
*Clavispora lusitaniae*	1	0	0	0

**Table 23 antibiotics-14-00561-t023:** Most common pathogens tested for vancomyvin resistance stratified by sensitivity.

Pathogen	Sensitivity	n
*Staphylococcus epidermidis*	S	1589
*Enterococcus faecium*	S	976
*Enterococcus faecium*	R	383
*Enterococcus faecalis*	S	528
*Staphylococcus hominis*	S	306
*Staphylococcus hominis*	R	1
*Staphylococcus capitis*	S	219
*Staphylococcus haemolyticus*	S	161
*Staphylococcus aureus*	S	115
*Streptococcus constellatus*	S	59

**Table 24 antibiotics-14-00561-t024:** Linear regression of the proportion of ineffective vancomycin consumption on time and total consumption. “Mibi Time” refers to the time of the microbiological examination in years from the start of the study period and “Total Consumption” refers to an estimated DDD.

Characteristic	Beta	95% CI	*p*-Value
(Intercept)	0.120	0.109, 0.132	0.000
Mibi Time	0.013	0.012, 0.014	0.000
Total Consumption	−0.031	−0.038, −0.023	0.000

CI = Confidence interval.

**Table 25 antibiotics-14-00561-t025:** Summary table of epidemiological characteristics of vancomycin-treated patients stratified by observed sensitivity.

Characteristic	n.a. N = 219	R N = 75	S N = 393	*p*-Value
Sex				0.015
female	103 (47%)	35 (47%)	142 (36%)	
male	116 (53%)	39 (53%)	251 (64%)	
unknown	0	1	0	
Age	66 (53, 76)	64 (57, 72)	64 (54, 74)	0.6
Destination				0.067
death	89 (41%)	28 (37%)	130 (33%)	
external hospital	71 (32%)	27 (36%)	174 (44%)	
home	59 (27%)	20 (27%)	89 (23%)	
ICU Duration	6 (2, 12)	9 (3, 22)	13 (4, 23)	<0.001
Hospital Duration	18 (6, 32)	51 (33, 87)	30 (16, 50)	<0.001

n (%); Median (Q1, Q3); Pearson’s chi-squared test; Kruskal–Wallis rank sum test.

**Table 26 antibiotics-14-00561-t026:** List of the 23 antimicrobials given to the patient with the most administrations during their hospital stay.

Ertapenem i.v.	Valganciclovir p.o.	Colistin i.v.
Fluconazol i.v.	Piperacillin/Tazobactam i.v.	Meropenem i.v.
Ganciclovir i.v.	Linezolid i.v.	Gentamicin i.v.
Metronidazol i.v.	Micafungin i.v.	Ceftolozan/Tazobactam i.v.
Caspofungin i.v.	Tigecyclin i.v.	Tobramycin i.v.
Imipenem/Cilastatin i.v.	Flucloxacillin i.v.	Voriconazol p.o.
Erythromycin i.v.	Cefepim i.v.	Ampicillin/Sulbactam i.v.
Vancomycin i.v.	Ciprofloxacin i.v.	

**Table 27 antibiotics-14-00561-t027:** Summary table of performed ampicillin/sulbactam tests stratified by observed sensitivity class.

Antibiotic	Sensitivity	n
Ampicillin		1
Ampicillin	I	510
Ampicillin	R	6455
Ampicillin	S	1274
Ampicillin/Sulbactam	I	1085
Ampicillin/Sulbactam	R	3081
Ampicillin/Sulbactam	S	910

**Table 28 antibiotics-14-00561-t028:** Most common pathogens tested for ampicillin/sulbactam resistance stratified by observed sensitivity.

Pathogen	Sensitivity	n
*Escherichia coli*	S	385
*Escherichia coli*	I	474
*Escherichia coli*	R	1329
*Klebsiella pneumoniae*	S	249
*Klebsiella pneumoniae*	I	235
*Klebsiella pneumoniae*	R	449
*Enterobacter cloacae*	R	420
*Proteus mirabilis*	S	87
*Proteus mirabilis*	I	165
*Proteus mirabilis*	R	49
*Serratia marcescens*	R	293

**Table 29 antibiotics-14-00561-t029:** Linear regression of the proportion of ineffective ampicillin/sulbactam consumption on time and total consumption. “Mibi Time” refers to the time of the microbiological examination in years from the start of the study period and “Total Consumption” refers to an estimated DDD.

Characteristic	Beta	95% CI	*p*-Value
(Intercept)	0.272	0.253, 0.291	0.000
Mibi Time	0.009	0.007, 0.011	0.000
Total Consumption	0.028	0.016, 0.040	0.000

CI = Confidence interval.

**Table 30 antibiotics-14-00561-t030:** Summary table of epidemiological characteristics of ampicillin/sulbactam-treated patients stratified by observerd sensitivity.

Characteristic	I N = 77	n.a. N = 449	R N = 292	S N = 85	*p*-Value
Sex					0.5
female	35 (45%)	185 (41%)	108 (37%)	32 (38%)	
male	42 (55%)	264 (59%)	184 (63%)	53 (62%)	
Age	71 (56, 79)	67 (53, 77)	68 (58, 77)	66 (57, 75)	0.4
Destination					0.002
death	14 (18%)	66 (15%)	55 (19%)	23 (27%)	
external hospital	34 (44%)	143 (32%)	115 (39%)	31 (36%)	
home	29 (38%)	240 (53%)	122 (42%)	31 (36%)	
ICU Duration	5 (3, 12)	3 (1, 7)	7 (3, 15)	8 (2, 15)	<0.001
Hospital Duration	21 (13, 38)	14 (9, 25)	28 (15, 49)	24 (12, 39)	<0.001

n (%); Median (Q1, Q3); Pearson’s chi-squared test; Kruskal–Wallis rank sum test.

**Table 31 antibiotics-14-00561-t031:** Summary table of epidemiological characteristics of Linezolid-treated patients stratified by observed sensitivity.

Characteristic	n.a. N = 103	R N = 24	S N = 244	*p*-Value
Sex				>0.9
female	42 (41%)	9 (38%)	102 (42%)	
male	61 (59%)	15 (63%)	141 (58%)	
unknown	0	0	1	
Age	66 (54, 76)	62 (49, 73)	65 (55, 75)	0.6
Destination				0.6
death	44 (43%)	8 (33%)	85 (35%)	
external hospital	34 (33%)	9 (38%)	83 (34%)	
home	25 (24%)	7 (29%)	76 (31%)	
ICU Duration	6 (2, 15)	15 (2, 49)	11 (4, 22)	0.009
Hospital Duration	25 (12, 42)	62 (30, 89)	41 (23, 62)	<0.001

n (%); Median (Q1, Q3); Pearson’s chi-squared test; Kruskal–Wallis rank sum test.

**Table 32 antibiotics-14-00561-t032:** Linear regression of the proportion of ineffective linezolid consumption on time and total consumption. “Mibi Time” refers to the time of the microbiological examination in years from the start of the study period and “Total Consumption” refers to an estimated DDD.

Characteristic	Beta	95% CI	*p*-Value
(Intercept)	0.048	0.031, 0.065	0.000
Mibi Time	0.002	0.000, 0.005	0.087
Total Consumption	0.057	0.046, 0.069	0.000

CI = Confidence interval.

**Table 33 antibiotics-14-00561-t033:** Linear regression of the proportion of ineffective ciprofloxacin consumption on time and total consumption. “Mibi Time” refers to the time of the microbiological examination in years from the start of the study period and “Total Consumption” refers to an estimated DDD.

Characteristic	Beta	95% CI	*p*-Value
(Intercept)	0.054	0.025, 0.083	0.000
Mibi Time	0.005	0.001, 0.010	0.010
Total Consumption	0.200	0.181, 0.219	0.000

CI = Confidence interval.

**Table 34 antibiotics-14-00561-t034:** The 10 most frequent pathogens isolated in patients treated with fluoroquinolones.

Pathogen	n
*Staphylococcus epidermidis*	969
*Candida albicans*	884
*Escherichia coli*	648
*Enterococcus faecium*	505
*Candida glabrata*	433
*Pseudomonas aeruginosa*	419
*Klebsiella pneumoniae*	383
*Staphylococcus aureus*	380
*Enterobacter cloacae*	238
*Serratia marcescens*	165

**Table 35 antibiotics-14-00561-t035:** Frequency of observed fluoroquinolone sensitivity classes.

Sensitivity Class	n
n.a.	3
I	295
R	1370
S	2065

**Table 36 antibiotics-14-00561-t036:** Summary table of epidemiological characteristics of ciprofloxacin-treated patients stratified by observed sensitivity.

Characteristic	I N = 37	n.a. N = 135	R N = 109	S N = 266	*p*-Value
Sex					0.080
female	18 (49%)	64 (47%)	42 (39%)	94 (35%)	
male	19 (51%)	71 (53%)	67 (61%)	172 (65%)	
Age	64 (57, 78)	68 (54, 77)	65 (54, 74)	69 (56, 77)	0.2
Destination					0.6
death	7 (19%)	40 (30%)	28 (26%)	62 (23%)	
external hospital	20 (54%)	56 (41%)	53 (49%)	135 (51%)	
home	10 (27%)	39 (29%)	28 (26%)	69 (26%)	
ICU Duration	14 (6, 23)	7 (1, 12)	14 (4, 28)	10 (5, 20)	<0.001
Hospital Duration	32 (22, 49)	18 (9, 31)	38 (21, 62)	28 (16, 41)	<0.001

n (%); Median (Q1, Q3); Pearson’s chi-squared test; Kruskal–Wallis rank sum test.

**Table 37 antibiotics-14-00561-t037:** Linear regression of the proportion of ineffective cefuroxim consumption on time and total consumption. “Mibi Time” refers to the time of the microbiological examination in years from the start of the study period and “Total Consumption” refers to an estimated DDD.

Characteristic	Beta	95% CI	*p*-Value
(Intercept)	0.280	0.254, 0.306	0.000
Mibi Time	−0.019	−0.024, −0.013	0.000
Total Consumption	−0.188	−0.223, −0.152	0.000
Mibi Time × Total Consumption	0.034	0.027, 0.041	0.000

CI = Confidence interval.

**Table 38 antibiotics-14-00561-t038:** Summary table of epidemiological characteristics stratified by observed sensitivity with respect to all 8 main antimicrobial representatives.

Characteristic	I N = 158	n.a. N = 1492	R N = 391	S N = 473	*p*-Value
Sex					0.6
female	63 (40%)	593 (40%)	152 (39%)	203 (43%)	
male	95 (60%)	898 (60%)	238 (61%)	270 (57%)	
unknown	0	1	1	0	
Age	73 (60, 81)	66 (55, 77)	68 (59, 77)	69 (60, 78)	<0.001
Destination					<0.001
death	38 (24%)	290 (19%)	87 (22%)	134 (28%)	
external hospital	78 (49%)	469 (31%)	143 (37%)	196 (41%)	
home	42 (27%)	733 (49%)	161 (41%)	143 (30%)	
ICU Duration	6 (2, 15)	3 (1, 8)	5 (2, 14)	5 (2, 12)	<0.001
Hospital Duration	19 (12, 33)	16 (9, 27)	28 (15, 47)	21 (11, 34)	<0.001

n (%); Median (Q1, Q3); Pearson’s chi-squared test; Kruskal–Wallis rank sum test.

## Data Availability

Data are available on request due to legal and ethical restrictions.

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
