# Peer review of "The Actual Clinical Situation Ruthlessly Exposes the Challenge of Rational Care for Nosocomial and Community-Acquired Infections and Requires Even More Efforts for Satisfactory Antibiotic Stewardship"

_antibiotics, 2025, doi:10.3390/antibiotics14060561_

Round 1
Reviewer 1 Report
Comments and Suggestions for Authors
Title
The title does not predict the content of the study. Possibly, a sentence such as: "First steps toward harnessing the complex dynamics of antibiotic resistance" is closer to the content of the study.
Introduction
Bibliographic citations should be introduced to support comments made in the introduction prior to citation 1.
The introduction is too long and should be summarized without going into detail about some of the accepted strategies in the field of antimicrobial stewardship.
Please clarify, could be 7,718 patients with only 9,201 ward stays were identified?
Line 314, MIC or cutt off?
Line 337, It is not clear that the formula can be extrapolated to the usual assessments of DDDs or DOTs. There is a lack of validation that makes this formula more understandable.
Results
The length of hospital stay in the ICU is striking, as shown in both Tables 1 and 2.
There is no adjustment for the severity of the patients, so the mortality described is questionable.
Global
The design of the study is questionable in order to make relationships between antibiotic use and the emergence of bacterial resistance, as well as relationships with mortality. As the authors comment, the limitations are very important, the study has a very complex interpretation. I would recommend a validation of the model against the usual practice used to determine the relationships between antibiotic consumption, development of resistance and mortality. On the other hand, the authors could deepen their model by comparing it with the standards used for antibiotic consumption expressed in DDDs or DOTs. There are other factors, beyond the consumption of antibiotics, that go unnoticed in the study design.
Comments on the Quality of English Languagenone
Author Response
Title
Comment 1:
The title does not predict the content of the study. Possibly, a sentence such as: "First steps toward harnessing the complex dynamics of antibiotic resistance" is closer to the content of the study.
Response 1:
Antibiotic stewardship is generally taken seriously in most German hospitals. We therefore initially assumed that the first steps had already been taken earlier and that the fine-tuning should now begin. The title is intended to express our surprise that the clinical situation presents considerably more complex and that a fundamental descriptive evaluation of the current situation must first be carried out. We sincerely believe that the chosen title expresses well the necessity that the efforts to establish a functioning ABS still need to be increased considerably. However, we added “community-acquired” to the title since we do not exclusively focus on nosocomial infections.
Introduction
Comment 2:
Bibliographic citations should be introduced to support comments made in the introduction prior to citation 1.
Response 2:
We agree that some references were missing. Accordingly, we added citations.
Comment 3:
The introduction is too long and should be summarized without going into detail about some of the accepted strategies in the field of antimicrobial stewardship.
Response 3:
We agree. We shortened the introduction accordingly.
Comment 4:
Please clarify, could be 7,718 patients with only 9,201 ward stays were identified?
Response 4:
We added a clarification to the text, namely that some patients had multiple hospitalizations.
Comment 5:
Line 314, MIC or cutt off?
Response 5:
We added a clarification “… sensitivity was determined by applying the threshold values to the MIC ...”
Comment 6:
Line 337, It is not clear that the formula can be extrapolated to the usual assessments of DDDs or DOTs. There is a lack of validation that makes this formula more understandable.
Response 6:
We agree that this aspect was communicated very poorly. The formula itself has nothing to do with the calculation of the DDD variant. Rather, the “standardization” is already included in the consumption densities A used in the formula. We have now described this in detail and precisely.
Results
Comment 7:
The length of hospital stay in the ICU is striking, as shown in both Tables 1 and 2.
Response 7:
After a check, we can confirm the figures given. The peculiarity perceived by the reviewer is not apparent to us.
Comment 8:
There is no adjustment for the severity of the patients, so the mortality described is questionable.
Response 8:
Estimating severity from available data is not straightforward. We agree, however, that this is an interesting point, although a causal approach is not feasible within the scope of an observational study, anyway. However, instead of simply adding severity as potential confounder, considering an interaction with AB treatment would indeed be informative. Nevertheless, an expected positive correlation between severity and treatment would inform us about the actual mechanism, but would not, or only slightly, correct the correlation between mortality and AB treatment. Nevertheless, this is a limitation that we add, and we will suggest an analysis of this aspect in future studies.
Global
Comment 9:
The design of the study is questionable in order to make relationships between antibiotic use and the emergence of bacterial resistance, as well as relationships with mortality. As the authors comment, the limitations are very important, the study has a very complex interpretation. I would recommend a validation of the model against the usual practice used to determine the relationships between antibiotic consumption, development of resistance and mortality. On the other hand, the authors could deepen their model by comparing it with the standards used for antibiotic consumption expressed in DDDs or DOTs. There are other factors, beyond the consumption of antibiotics, that go unnoticed in the study design.
Response 9:
The study is designed as an exploratory study. It is important to note at this point that the German ABS guidelines only require quarterly aggregated registration of antibiogram results at the Robert Koch Institute (RKI) (the central federal institution responsible for disease prevention and control in Germany). The availability of data at the individual level, which we are fortunately able to access, is a special feature of the clinic under investigation. In other words, we are already far ahead of the usual practice of evaluating the incidence of resistance in Germany and correlate it with mortality. Nevertheless, this is a retrospective study, i.e. in order to be able to address the questions proposed by the reviewer in a sound manner, prospective planning is required. It is our concern and our great hope that we can encourage prospective studies designed specifically for these questions. Furthermore, the present work represents a first attempt to provide a basis for more in-depth evaluations. We therefore ask the reviewer to consider and accept the descriptive results of this exploratory study essentially as a hypothesis generation for further in-depth analysis.
Reviewer 2 Report
Comments and Suggestions for Authors
The authors have made a commendable effort in understanding the complex dynamics between various factors governing development of antibiotic resistance and chalking out a proposed action plan to address the hazard of antibiotic resistance. The manuscript is very comprehensively written and puts forth various issues quite explicitly. There are some concerns, however, which need to be addressed:
Introduction:
The authors have explained antibiotic cycling or mixing and scheduled cycling as strategies under AMS. I would advise authors to add few lines on how these strategies differ from each other in terms of implementation and antibiotic resistance outcomes. Below is a review article addressing this topic:
Beardmore RE, Peña-Miller R, Gori F, Iredell J. Antibiotic Cycling and Antibiotic Mixing: Which One Best Mitigates Antibiotic Resistance? Mol Biol Evol. 2017 Apr 1;34(4):802-817. doi: 10.1093/molbev/msw292. PMID: 28096304; PMCID: PMC5400377.
Methods:
Some more details in methodology including study setting, period of study conduct, data retrieval and the database of electronic health records may be added to make it more understandable to the readers and enhance the robustness and reproducibility.
Please mention full form of SQL (SQL based data query).
Since the title mentions “nosocomial infections”, it is worth mentioning which types of nosocomial infections were the focus of interest in the present study? Standard case definitions of the included nosocomial infections should also be added.
Results
Table 1: what is meant by: gt median, leq median?
Along with the demographic data of patients, clinical characteristics such as admission diagnosis, presence of any indwelling device (ET tube, central line, catheter etc.), APACHE score etc. should be added in view of critically ill patients. Also, from which type of ICUs was the data collected? Was the data collected from adults only or also pediatric patients?
Table 2: the stratification of antibiotic treatment flag also needs to be studied with respect to clinical characteristics as co-variates.
The authors have studied series of uninterrupted daily antibiotic treatments per patient. Please provide any literature evidence for the criteria used, if any.
Table 5: sensibility should be corrected to “sensitivity” or “susceptibility”.
For antibiogram under section 3.1.4, what were the sites of culture specimens viz. blood, urine, pus, endotracheal aspirate etc.
The authors have performed many linear regression analyses to study correlation between various co-variates and resistance. It is also important to study the correlation between number of antibiotics per patient and resistance which was probably missed out. Also, resistance patterns in different types of nosocomial infections should be studied.
Discussion and conclusion
Additional data and analyses pertaining to nosocomial infections per se should be provided as mentioned above and to be discussed along with the strategies to address them.
Comments on the Quality of English LanguageEnglish language needs moderate corrections.
Author Response
The authors have made a commendable effort in understanding the complex dynamics between various factors governing development of antibiotic resistance and chalking out a proposed action plan to address the hazard of antibiotic resistance. The manuscript is very comprehensively written and puts forth various issues quite explicitly. There are some concerns, however, which need to be addressed:
Introduction:
Comment 1:
The authors have explained antibiotic cycling or mixing and scheduled cycling as strategies under AMS. I would advise authors to add few lines on how these strategies differ from each other in terms of implementation and antibiotic resistance outcomes. Below is a review article addressing this topic:
Beardmore RE, Peña-Miller R, Gori F, Iredell J. Antibiotic Cycling and Antibiotic Mixing: Which One Best Mitigates Antibiotic Resistance? Mol Biol Evol. 2017 Apr 1;34(4):802-817. doi: 10.1093/molbev/msw292. PMID: 28096304; PMCID: PMC5400377.
Response 1:
In fact, we have neglected to refer to this fundamental work. We thank the reviewer for the reference. We have now included the according citation. We have already given brief explanations of the difference and significance of the various strategies in the first version and we do not want to go beyond the scope of the introduction here, since Reviewer 1 even asked us to shorten the introduction, so we will be brief here.
Methods:
Comment 2:
Some more details in methodology including study setting, period of study conduct, data retrieval and the database of electronic health records may be added to make it more understandable to the readers and enhance the robustness and reproducibility.
Response 2:
We added a subsection on the study type within Materials and Methods to clarify the setting (new section 2.1). Additionally, we made a few adaptations in subsequent sections. However, some aspects mentioned by the reviewer concerning the conduct of the study and the observation period are already included in the first version. A public release of the data is not legally intended. We assume that the content and structure of the dataset are sufficiently well described in the manuscript. We also refer to the following responses.
Comment 3:
Please mention full form of SQL (SQL based data query).
Response 3:
We fully agree with the reviewer that the provision of the analyzed data is in principle desirable and represents an important momentum in terms of reproducibility and assessability of the research. However, it is also known that authors often have to strike a balance between the demand for open research and legal or regulatory requirements. The German regulations certainly have a special status in this respect, as the obligation to use data in a patient-oriented manner and the sovereignty of such data are often in competition. The provision of data for scientific purposes is often subject to the vested interests and policies of decision-makers, rather than ethical imperatives. Specifically in the context of antibiotic resistance, the only obligation in Germany to date is that only aggregated data on consumption and observed resistance must be delivered to a surveillance program (operated by the Robert Koch Institute, RKI, the central federal institution responsible for disease prevention and control in Germany) on a quarterly basis. All experts are constantly calling for data with a higher temporal resolution and, if possible, even individual data, but unfortunately there is still no centralized form of data acquisition. By evaluating the clinical data available to us, we want to exert pressure for the necessary standardization, but we cannot and do not want to disregard the current regulations. We therefore appeal to the reviewer and the editors to be satisfied with what we consider to be a sufficiently good description of the data set. We also emphasize once again that this is an exploratory retrospective study, which also has the character of a feasibility study with regard to prospective design. In the latter case, we consider the provision of the raw data to be obligatory.
Comment 4:
Since the title mentions “nosocomial infections”, it is worth mentioning which types of nosocomial infections were the focus of interest in the present study? Standard case definitions of the included nosocomial infections should also be added.
Response 4:
Admittedly, we have not used the term nosocomial in a very well-founded way. We have adapted both the title and the corresponding explanations in the text and added “community-acquired”. In fact, the data set analyzed does not primarily distinguish between community-acquired and nosocomial infections. The pathogens that we focus on in the evaluation are those that cause massive problems, especially in hospital-acquired infections, but since the focus here is on the development of resistance to treatment, we have initially refrained from separating the two origins of infection.
Results
Comment 5:
Table 1: what is meant by: gt median, leq median?
Response 5:
gt refers to greater than, i.e. >, and leq means less or equal, i.e. <=. We changed the nomenclature accordingly.
Comment 6:
Along with the demographic data of patients, clinical characteristics such as admission diagnosis, presence of any indwelling device (ET tube, central line, catheter etc.), APACHE score etc. should be added in view of critically ill patients. Also, from which type of ICUs was the data collected? Was the data collected from adults only or also pediatric patients?
Response 6:
The analyzed ICU is an operational intensive care unit that does not generally include the care of pediatric patients, but in very rare cases emergency care may be provided for newborns or children who remain in the ICU until stable transport is possible. These patients do not play a role in the analysis presented here. Please refer also to the new subsection 2.1 where the analysed ICU is now described in detail.
The consideration of additional covariates is planned in a future publication. Deriving severity scores from the countless laboratory values and other clinical and diagnostic data is far beyond the scope of the descriptive and exploratory approach adopted. The presentation of the myriads of observed and recorded clinical data, including the collection sites of the isolates, would make the presentation completely confusing without really contributing anything substantial to this overview study. The sheer volume of covariates requires an elaborate analysis strategy based on pattern recognition using modern machine learning algorithms. The “Outlook” section contains references to corresponding suggestions. We consider the reviewers' suggestions to be absolutely important and we are doing everything we can to take these suggestions on board in the further analysis of the available data. However, we are also absolutely convinced that this work is a necessary basis for developing the next steps. In our opinion, the work also provides a basis for a relevant community and we consider it to be a guideline for further efforts, especially for the German research landscape.
Comment 7:
Table 2: the stratification of antibiotic treatment flag also needs to be studied with respect to clinical characteristics as co-variates.
Response 7:
This point is closely related to response 6. We therefore ask the reviewer to also consider this point in the light of our previous response.
Comment 8:
The authors have studied series of uninterrupted daily antibiotic treatments per patient. Please provide any literature evidence for the criteria used, if any.
Response 8:
The rationale behind this approach follows from an approach used in compartmental modeling of infection epidemiological processes. In this context, the term SIR models is often used for short. Individual consumption, infections and resistance proportions are aggregated to form corresponding density functions. This approach makes it possible to use the repertoire of methods from the field of mathematical dynamic modeling. This includes the presented mutual correlations and delay correlations, as well as the presented phase space representations. Reference was made to a corresponding publication that evaluates such a procedure. The approach also makes it possible to make the mathematical methodology that we recently published usable in the future (10.1371/journal.pone.0238692). Please see the changes made in section 2.4.2.
Comment 9:
Table 5: sensibility should be corrected to “sensitivity” or “susceptibility”.
Response 9:
We changed it consistently throughout the manuscript.
Comment 10:
For antibiogram under section 3.1.4, what were the sites of culture specimens viz. blood, urine, pus, endotracheal aspirate etc.
Response 10:
We thank the reviewer for pointing out that the sites where the samples were taken must be mentioned. In total, isolates were taken from 41 different sample types. However, some were unspecific, e.g. smear, and some were more differentiated, e.g. ulcer smear. Sample types with frequencies above 1,000 each are with decreasing frequency: tracheal secretion, urine, anaerobic blood culture, aerobic blood culture, deep swab, bronchial lavage. Evaluations with sample type as a predictor are not considered in this descriptive review.
Comment 11:
The authors have performed many linear regression analyses to study correlation between various co-variates and resistance. It is also important to study the correlation between number of antibiotics per patient and resistance which was probably missed out. Also, resistance patterns in different types of nosocomial infections should be studied.
Response 11:
Section 3.1.5 does no contain the regression result that includes the number of administered antibiotics as predictor. We thank the reviewer for the very valuable hint that we initially forgot to evaluate the effect of this important predictor.
Discussion and conclusion
Comment 12:
Additional data and analyses pertaining to nosocomial infections per se should be provided as mentioned above and to be discussed along with the strategies to address them.
Response 12:
We apologize again that the wording gave the wrong impression that the origin of the infections was addressed. However, the reviewer's comments are very welcome. We are not only willing, but have also firmly planned to evaluate the necessary strata and subgroups in further studies. Once again, we would like to emphasize that the present study is a first, but nonetheless an extremely important step in the preparation of such follow-up studies.
Comment 13:
English language needs moderate corrections.
Response 13:
We went through the manuscript for linguistic irregularities and made the necessary corrections.
Round 2
Reviewer 1 Report
Comments and Suggestions for Authors
The study has been presented in a more understandable form and represents a new line of research to explore.
Reviewer 2 Report
Comments and Suggestions for Authors
The authors have addressed all the queries. No further comments from my side.